# Prdm1 positively regulates liver Group 1 ILCs cancer immune surveillance and preserves functional heterogeneity

Jitian He[1,2†], Le Gao[1†], Peiying Wang[1†], Wing Keung Chan[3], Yiran Zheng[1], Yumo Zhang[1], Jiaman Sun[1], Xue Li[4], Jiming Wang[5], Xiao-Hong Li[1*‡], Huaiyong Chen[4,6,7*‡], Zhouxin Yang[8*‡], Youwei Wang[1*‡]

[1]Institute of Medical Engineering & Translational Medicine, Tianjin University, Tianjin, China; [2]Organ Transplant Center, The First Affiliated Hospital of Sun Yat-sen University, Guangzhou, China; [3]Department of Internal Medicine, Division of Hematology, The Ohio State University, Columbus, United States; [4]Department of Basic Medicine, Haihe Hospital, Tianjin University, Tianjin, China; [5]Tianjin Economic-Technological Development Area (TEDA) Hospital, Tianjin, China; [6]College of Pulmonary and Critical Care Medicine, 8th Medical Center, Chinese PLA General Hospital, Beijing, China; [7]Tianjin Key Laboratory of Lung Regenerative Medicine, Tianjin, China; [8]Zhejiang Provincial Key Lab of Geriatrics and Geriatrics Institute of Zhejiang Province, Zhejiang Hospital, Hangzhou, China

*For correspondence:
xhli18@tju.edu.cn (X-HL);
huaiyong.chen@foxmail.com
(HC);
yangzhouxin@hotmail.com (ZY);
youwei.wang@tju.edu.cn (YW)

†These authors contributed equally to this work
‡These authors also contributed equally to this work

Competing interest: The authors declare that no competing interests exist.

**Abstract** Group 1 innate lymphoid cells (ILCs) comprise conventional natural killer (cNK) cells and type 1 innate lymphoid cells (ILC1s). The main functions of liver cNK cells and ILC1s not only include directly killing target cells but also regulating local immune microenvironment of the liver through the secretion of cytokines. Uncovering the intricate mechanisms by which transcriptional factors regulate and influence the functions of liver cNK cells and ILC1s, particularly within the context of liver tumors, presents a significant opportunity to amplify the effectiveness of immunotherapies against liver malignancies. Using Ncr1-driven conditional knockout mouse model, our study reveals the regulatory role of *Prdm1* in shaping the composition and maturation of cNK cells. Although *Prdm1* did not affect the killing function of cNK cells in an in vivo cytotoxicity model, a significant increase in cancer metastasis was observed in *Prdm1* knockout mice. Interferon-gamma (IFN-γ), granzyme B, and perforin secretion decreased significantly in *Prdm1*-deficient cNK cells and liver ILC1s. Single-cell RNA sequencing (scRNA-seq) data also provided evidences that Prdm1 maintains functional subsets of cNK cells and liver ILC1s and facilitates communications between cNK cells, liver ILC1s, and macrophages. The present study unveiled a novel regulatory mechanism of Prdm1 in cNK cells and liver ILC1s, showing promising potential for developing innovative immune therapy strategies against liver cancer.

## eLife assessment

The authors investigated the requirement and function of Blimp1/Prdm1 in murine natural killer (NK) cells and the ILC1 lineage of innate lymphoid cells, using a conditional knockout model. The single-cell mRNA-seq data provided here represent a **valuable** resource for the community, but the lack of mechanistic investigations leaves the study partially **incomplete**. The work will be of interest to the fields of innate lymphoid cell biology and tissue immunology.

## Introduction

Group 1 ILCs consist of cNK cells and ILC1s (*Spits et al., 2013*; *Peng et al., 2013*), with distinct developmental trajectories and effect molecules (*Bai et al., 2021*). Both cNK cells and ILC1s play indispensable roles in combatting viral infections (*Weizman et al., 2017*), maintaining local immune homeostasis (*Schuster et al., 2014*), eradicating malignant transformed cells (*Ducimetière et al., 2021*), and fostering cross-talk with adaptive immunity (*Fumagalli et al., 2022*). cNK cells demonstrate potent cellular cytotoxicity, facilitating direct elimination of target cells (*Kiessling et al., 1975*). On the contrary, a defining attribute of ILC1s is their predominant cytokine-mediated functions, with limited cellular killing capacity (*Nabekura et al., 2020*). In a state of homeostasis, liver group 1 ILCs (CD45$^+$CD3$^-$NK1.1$^+$NKp46$^+$) can be discriminated into cNK cells and ILC1s by the differential expression of CD49a and CD49b (*Peng et al., 2013*): cNK cells are marked by the expression of CD49b, while liver ILC1s exhibit a distinctive positivity for CD49a. Tumor Necrosis Factor Related Apoptosis Inducing Ligand (TRAIL) is also expressed on liver ILC1s, but not on cNK cells (*Takeda et al., 2001*). Transcriptional factors (TFs) such as T-bet, Nfil3, PLZF, and ID2 are required for cNK cells and liver ILC1s development or generation of their progenitors (*Townsend et al., 2004*; *Yokota et al., 1999*; *Gascoyne et al., 2009*; *Constantinides et al., 2014*). Eomes is considered to be necessary for NK cell maturation, but not for ILC1s (*Gordon et al., 2012*). Liver environment facilitated T-bet expression in the early stage of NK cells development, which results in Eomes repression. The repression of T-bet is required for Eomes$^+$ NK cells (*Daussy et al., 2014*). On the contrary, deficiency of Hobit results in the depletion of ILC1s, while only leaving little impact on liver NK cells (*Mackay et al., 2016*). These studies suggest that some TFs may, or usually, play different roles in NK cells and ILC1s.

During the development of liver cancers, the functions of immune cells are often inhibited, resulting in the formation of an immunosuppressive tumor microenvironment, and sometimes even systemic immune suppression (*Lee et al., 2022*). Transforming a poorly immunogenic tumor into a immunologically 'hot' tumor with a stronger immune response is one of the goals of many cancer immunotherapies (*Galon and Bruni, 2019*). Research targeting the immune system is showing increasing clinical promise in liver cancer treatment (*Ruff et al., 2022*). One of our recent studies also found that Toll-like receptor agonists can significantly enhance the anti-tumor effect of Sorafenib by reconstructing the tumor immune microenvironment and reshaping the vascular system (*He et al., 2023*). The anti-tumor activity of cNK cells in the liver is relatively well-established, whereas the relationship between liver ILC1 and tumors is still a topic of controversy. Clinical data showed that liver tumors with a higher infiltration of NK cells are associated with a better prognosis (*Nersesian et al., 2021*). Decreased NK cell activity is closely correlated with the malignancy of liver tumors and represents a significant risk factor for recurrence (*Lee et al., 2021*). The tumor microenvironment orchestrates the transformation of cNK cells into ILC1s in a TGF-β-dependent manner, resulting in their diminished capacity to control tumor growth and metastasis. This process ultimately promotes tumor immunoevasion (*Gao et al., 2017*). However, researches also revealed that, distinct from directly inhibiting tumor growth, the primary function of ILC1 is to suppress the seeding of metastatic tumor cells in liver tissue (*Ducimetière et al., 2021*). Comprehensive research in this field is essential to harness the precision of group 1 ILCs targeted immunotherapy for liver cancers.

The transcription factor network governs the function of group 1 ILCs and the balance between cNK and ILC1. Our research team, along with others, has observed that one of the transcription factor in TGF-β pathway, Smad4, promotes the shift in balance from cNK cells towards ILC1 in a TGF-beta-independent pathway, and simultaneously, it positively regulates the expression of another transcription factor, PR domain 1 (Prdm1/Blimp1; *Wang et al., 2018*; *Cortez et al., 2017*). Prdm1 plays a crucial role in the differentiation of B cells into plasma cells and the homeostasis of T cells (*Keller and Maniatis, 1991*; *Kallies et al., 2006*; *Martins et al., 2006*). The function and regulatory network of Prdm1 in NK cells is distinct from B cells and T cells, for its expression is independent of B-cell lymphoma 6 (Bcl-6) and Interferon Regulatory actor 4 (IRF4) but relies on T-bet. In the study that identified Prdm1 as an essential transcriptional factor for NK cell maturation (*Kallies et al., 2011*), no significant difference was observed in IFN-γ production or cytotoxicity between Prdm1-deficient and wild-type NK cells. Although Prdm1 expression is dependent on IL-15 in immature NK cells and can be further upregulated by IL-12 or IL-21, it plays a role in downregulating the expression of certain cytokine receptors, such as CD25 (IL-2Rα), consequently diminishing the responsiveness of NK cells to IL-2 (*Akman et al., 2021*). Notably, some studies even suggested that Prdm1 suppressed the secretion

of IFN-γ in NK cells (*Smith et al., 2010*). These findings imply that, although Prdm1 promotes NK cell development, it may function as a negative regulator of their activity. However, direct evidence supporting the impact of Prdm1 on NK cell anti-tumor capabilities is still lacking. Furthermore, there has been no investigation into its influence on the homeostasis of cNK cells and ILC1s in the liver, nor has there been an exploration of the underlying mechanisms.

In the current study, we found that the deletion of *Prdm1* in Ncr1+ cells resulted in an imbalance in the homeostasis of liver group 1 ILCs, with a shift towards cNK cells. The data also support the essential role of Prdm1 in the cancer surveillance mediated by cNK cells. While Prdm1 positively regulates genes associated with cellular cytotoxicity, it concurrently exerts inhibitory control over certain positive regulators of NK cell development and functionality, such as JunB. Using scRNA-seq, we have identified a subset of cNK cells within the liver characterized by elevated JunB expression. These cells exhibit decreased expression of genes associated with cellular cytotoxicity, and their abundance significantly increases following Prdm1 knockout. Furthermore, our data also support that Prdm1 promotes the cross-talk between group 1 ILCs and macrophages.

## Results

### *Prdm1* promotes group 1 ILCs homeostasis and terminal maturation

Examination of 363 liver hepatocellular carcinoma (LIHC) patient samples from The Cancer Genome Atlas (TCGA) revealed a positive correlation between the expression of NK-cell-associated genes (*Cursons et al., 2019*) (*NCR1*, *KLRB1*, *CD160*, *PRF1*, etc.) and *PRDM1* expression (*Figure 1A*). The patients are ordered from highest to lowest based on the expression of NK-PRDM1 for survival analysis (*Figure 1B*). Notably, patients exhibiting higher levels of NK-PRDM1 expression (above the median) experienced better survival outcomes compared to those with lower levels of NK-PRDM1 expression (below the median; *Figure 1C*). Similar results were also found in skin cutaneous melanoma (SKCM, n=454) and lung adenocarcinoma (LUAD, n=497) patients (*Figure 1—figure supplement 1A-F*). Patients within the highest quartile of NK-PRDM1 signature expression demonstrated enhanced overall survival, a result that achieved statistical significance in LUAD and SKCM patients (*Figure 1—figure supplement 1G-I*). These data suggested that *PRDM1* in NK cells might be essential for immune surveillance in solid tumors, including liver cancer, and prompted us to investigate the function and mechanism of *PRDM1* in NK cells and ILC1 within the context of liver cancer.

*Ncr1-Cre* mice were crossed with *Prdm1^fl/fl* mice to generate *Ncr1-cre Prdm1^fl/fl* mice, which specifically knockout exons 6–8 of *Prdm1* (*Shapiro-Shelef et al., 2003*) in NKp46-positive cells (*Figure 1D*). The mice carrying *Ncr1^Cre/+Prdm1^fl/fl* were referred to as *Prdm1* conditional knockout (*Prdm1* cko) mice, and the mice carrying *Ncr1^+/+Prdm1^fl/fl* were referred to as *Prdm1^+/+* mice. To further validate the deletion of *Prdm1* in NKp46+ cells of *Prdm1* cko mice, CD3-NK1.1+NKp46+ cells were sorted and the expression of *Prdm1* was measured by real-time RT-PCR. The expression of *Prdm1* was almost undetectable in CD3-NK1.1+NKp46+ cells of *Prdm1* cko mice, while was similar between total splenocytes in *Prdm1^+/+* and *Prdm1* cko mice (*Figure 1*, E and F) indicating successful, and specific knockout of *Prdm1* in NKp46+ cells.

Proportion and absolute number of NK cells in blood, bone marrow, lung, liver, spleen, and lymph nodes were analyzed by flow cytometry. Compared with *Prdm1^+/+* mice, the percentage and absolute number of NK cells (CD45+CD3-NK1.1+NKp46+) among lymphocytes was decreased in all of these tissues, whereas increased number of NK cells were observed in bone marrow (*Figure 1G*; *Figure 1—figure supplement 2A*). NK cell terminal maturation can be divided into four stages according to the expression of CD11b and CD27 (*Hayakawa and Smyth, 2006*). Stage IV (CD11b+CD27-) NK cells show higher level of effector molecules than any other stages and are considered as the most mature NK cells (*Chiossone et al., 2009*). The maturation of cNK cells (gated by CD45+CD3-NK1.1+NKp46+CD49b+) from blood, bone marrow, lung, liver, spleen, and lymph nodes were assessed, based on the expression of CD11b and CD27. Compared with *Prdm1^+/+* mice, the proportion of the most mature CD11b+CD27- NK cells were significantly decreased in all of the analyzed tissues in *Prdm1* cko mice (*Figure 1H*; *Figure 1—figure supplement 2B*). Killer cell lectin-like receptor subfamily G member 1 (KLRG1) is a lectin-like receptor which was considered as another marker of NK cell maturation (*Huntington et al., 2007*). In both *Prdm1^+/+* and *Prdm1* cko mice, NK cells from the liver and lung had the highest expression of KLRG1, following by blood and spleen (*Figure 1—figure supplement 2C*).

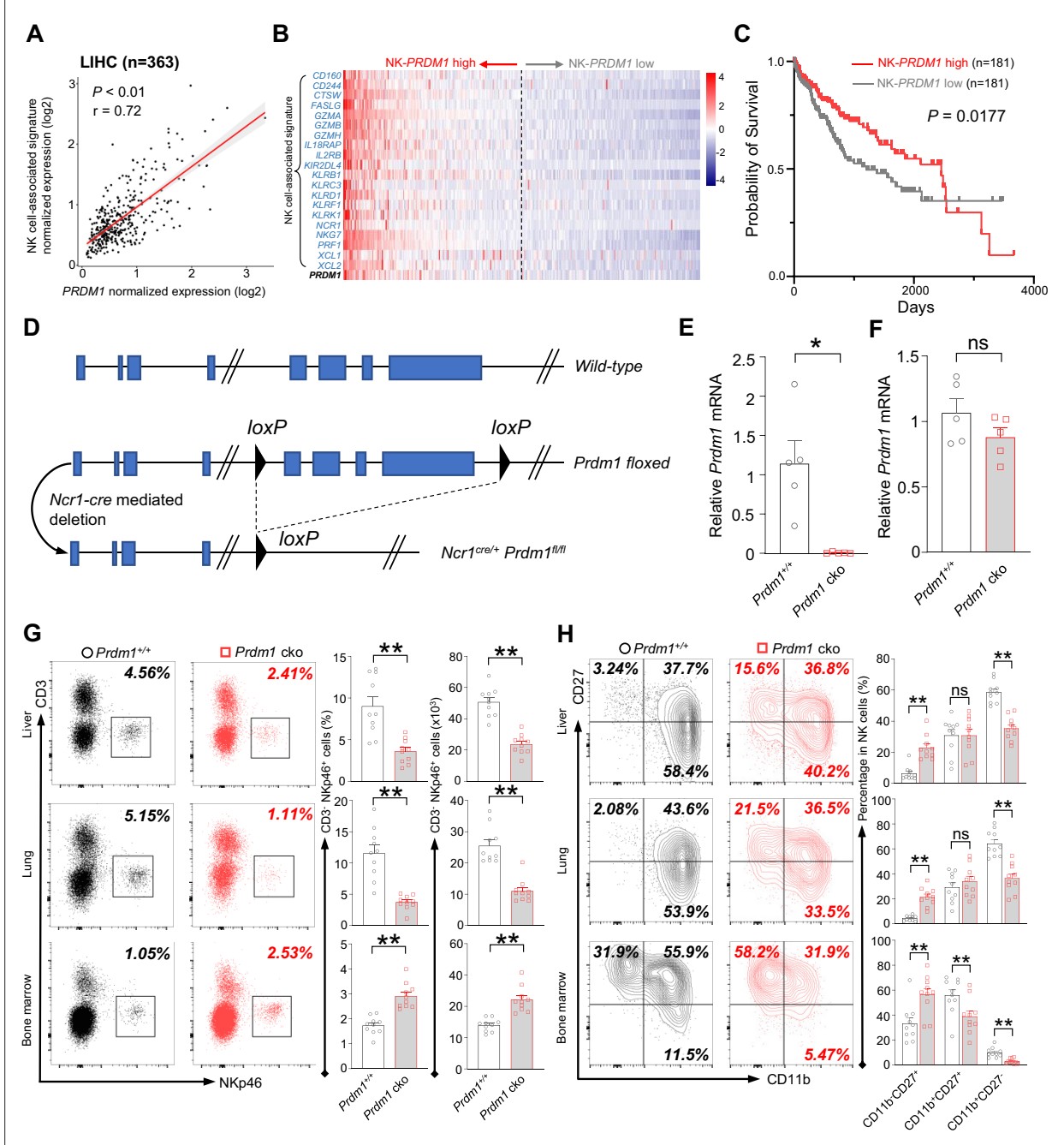

**Figure 1.** *Prdm1* promotes NK cell homeostasis and terminal maturation. (**A**) Correlation between the average expression of NK cell-associated signature and *PRDM1* in LIHC (Liver Hepatocellular Carcinoma; n=363) patients sourced from TCGA datasets. (**B**) Heatmap of the ordered, z-score normalized expression values for *PRDM1* and NK cell-associated genes in liver cancer patients. High and low expression of *NK-PRDM1* signature are indicated. (**C**) Prognostic value of the *NK-PRDM1* signature for overall survival of liver cancer patients comparing high and low samples with a median cutoff. (**D**) Schematic representation of the Prdm1-conditional knockout mouse model. Targeted exon 6–8 of *Prdm1* (top) is flanked with *loxP* sites (middle). Ncr1-expressed Cre recombinase was used to generate the *Prdm1* cko allele (bottom). (**E and F**) Real-time RT-PCR quantification of *Prdm1* expression in NKp46+ cells (**E**) and splenocytes (**F**) to determine the presence of *Prdm1* (n=5). (**G**) Representative flow cytometric plots (left) and quantification (right) of the proportion and absolute number of CD45+CD3-NKp46+ cells among lymphocytes in liver, lung, and bone marrow (n=10). (**H**) Representative flow cytometric plots (left) of the CD11b and CD27 expression within CD3-NK1.1+NKp46+CD49b+ cells in liver, lung, and bone marrow (n=10). Right panel showed the percentage of distinct maturation stages of NK cells. Data are presented as the mean ± SEM and were analyzed by two-tailed, paired t-test. Differences were evaluated between littermates. Each circle and square on graphs represents an individual mouse; *P*, p-value; r, pearson correlation coefficient; *, p<0.05; **, p<0.01, ns, not significant.

*Figure 1 continued on next page*

*Figure 1 continued*

The online version of this article includes the following figure supplement(s) for figure 1:

**Figure supplement 1.** High expression of *PRDM1-NK* signature predicts better overall survival of cancer patients.

**Figure supplement 2.** *Prdm1* plays an essential role in group 1 ILCs homeostasis and maturation.

The lowest KLRG1 expression was observed in NK cells derived from lymph nodes and bone marrow, indicating the presence of the most immature NK cells in these tissues. Consistent with CD11b/CD27 based maturation analysis, a significant loss of KLRG1$^+$ cells in NK cells was observed in *Prdm1* cko mice (*Figure 1—figure supplement 2C*). Together, these data confirmed that *Prdm1* is required for the terminal maturation of NK cells among various tissues.

## *Prdm1* is required for group 1 ILCs to control tumor metastasis

Two subpopulations of liver group 1 ILCs (gated by CD45$^+$CD3$^-$NK1.1$^+$NKp46$^+$) were further analyzed based on the expression of CD49a and CD49b (*Figure 2A*). Compared with *Prdm1*$^{+/+}$ mice, the *Prdm1* cko mice exhibited an increased percentage of cNK cells (CD49a$^-$CD49b$^+$) and reduced proportion of ILC1s (CD49a$^+$CD49b$^-$) (*Figure 2A*). Of note, the absolute number of both cNK cells and ILC1s were decreased in *Prdm1* cko mice, with a more robustly reduction in ILC1s (*Figure 2A*), which underscored the crucial role of *Prdm1* in maintaining the quantity of both liver cNK cells and ILC1s. Expression level of CD49b was slightly upregulated in *Prdm1* cko cNK cells and NKp46$^+$ cells in the liver and other tissues (*Figure 1—figure supplement 2D and E*). Increased CD49a expression was also observed in *Prdm1* cko liver ILC1s, while it showed decreased expression in NKp46$^+$ cells in the liver, bone marrow, and lymph nodes (*Figure 1—figure supplement 2F and G*). These results indicated the essential role of Prdm1 in maintaining the balance and hemostasis of cNK cells and ILC1s.

The decreased quantity of NK cell homeostasis and maturation of cNK cells due to *Prdm1* loss motivated us to further explore whether deficiency of Prdm1 impaired NK cell cytotoxicity. A B2M-deficient cell-based in vivo cytotoxicity assay was used to evaluate the effect of *Prdm1* on the cytotoxicity of NK cells (*Bix et al., 1991*; *Bern et al., 2019*). B2M-deficient cells do not have detectable Major Histocompatibility Complex I (MHC-I) on the cell surface, making them the target of NK cells (*Kärre et al., 1986*). Healthy NK cells will reject B2M-deficient donor cells efficiently and the elimination was used to quantify the cytotoxicity of NK cells. Although significant impaired homeostasis and maturation of NK cells were observed in *Prdm1* cko mice, no significant difference in the in vivo cytotoxicity assay were observed between *Prdm1*$^{+/+}$ and *Prdm1* cko mice (*Figure 1—figure supplement 2H and I*).

Besides their direct cytotoxic capabilities, NK cells' anti-tumor potential is also influenced by additional factors. These include their ability to counteract tumor-induced immune suppression and exhaustion, enhancing their effectiveness against cancer cells. Furthermore, NK cells can secrete cytokines that activate other immune cells, thereby orchestrating a broader immune response for the elimination of tumors. Moreover, the cytotoxicity assay, due to its relatively short duration, might not fully represent the anti-tumor activity of NK cells when continuously exposed to immune inhibitory signals in the tumor microenvironment. Therefore, we initiated an in vivo tumor model to further investigate the impact of *Prdm1* on NK cell anti-tumor capability. B16F10 is a melanoma cell line with low expression of MHC-I, which was susceptible to NK cell killing and usually used to evaluate NK cell anti-tumor capacity (*Viant et al., 2014*; *Cillo et al., 1987*; *Sathe et al., 2014*). The B16F10 cells were intravenously (for lung metastasis) or intrasplenic (for liver metastasis) administrated in the mice. The melanoma nodes were quantified 3 (intravenous injection) or 2 (intrasplenic injection) weeks after tumor inoculation. Compared with *Prdm1*$^{+/+}$ mice, deficiency of *Prdm1* resulted in more metastasis nodules in both lung (~twofold) and liver (~fourfold) (*Figure 2B and D*). Histological analysis further confirmed the increased frequency of metastasis tumor foci in *Prdm1* cko mice (*Figure 2C and E*). In agreement with the in vivo data, we also observed decreased IFN-γ secretion in *Prdm1* cko mice-derived splenic cNK cells, liver cNK cells, and liver ILC1s when stimulated by IL-18 alone or IL-12/IL-18 (*Figure 2F*; *Figure 2—figure supplement 1*; *Figure 7—figure supplement 1B*), which indicated that *Prdm1* is required for full activation of cNK cells and ILC1s in the context of IFN-γ production. These data implied that *Prdm1* is indispensable for NK-cell-mediated tumor surveillance.

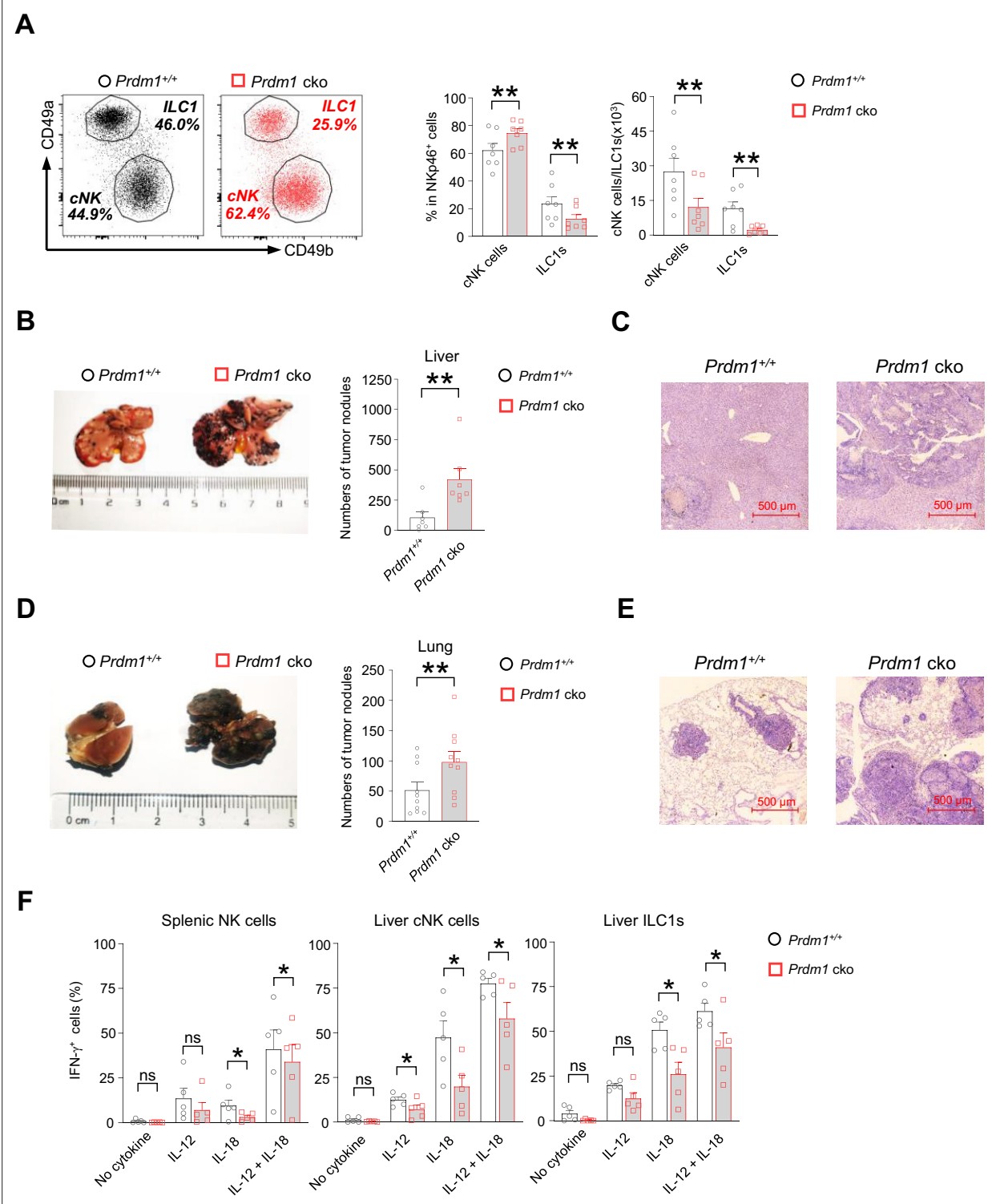

**Figure 2.** *Prdm1* is required for group 1 ILCs to control tumor metastasis. (**A**) Representative flow cytometric plots (left) of liver cNK cells (CD49a⁻ CD49b⁺) and ILC1s (CD49a⁺CD49b⁻) from *Prdm1*⁺/⁺ and *Prdm1* cko mice. The two bar graphs (right) quantitate the percentages and absolute numbers of cells respectively (n=7). (**B and D**) Representative image (left) and quantification (right) of tumor nodes on the livers (n=7) (**B**) and lungs (n=10) (**D**) of *Prdm1*⁺/⁺ and *Prdm1* cko mice at day 14 or 21 after inoculation with B16F10 melanoma cells. (**C and E**) Representative histopathological images of liver (**C**) and lung (**E**) tissues stained by hematoxylin-eosin to detect tumor metastasis. Red bar indicates 500 μm distance under the microscope. (**F**) Liver cells and splenocytes were co-stimulated in the presence or absence of IL-12 and IL-18 for 12 hr. GolgiStop was added 4 hr before intracellular staining of IFN-γ. The graphs showed percentage of IFN-γ⁺ splenic cNK cells, liver cNK cells and ILC1s from *Prdm1*⁺/⁺ and *Prdm1* cko mice (n=5). Data are

*Figure 2 continued on next page*

*Figure 2 continued*

presented as the mean ± SEM and were analyzed by two-tailed, paired t-test. Differences were evaluated between littermates. Each circle and square on graphs represents an individual mouse; *P*, p-value; *, p<0.05; **, p<0.01, ns, not significant.

The online version of this article includes the following figure supplement(s) for figure 2:

**Figure supplement 1.** *Prdm1* affects the IFN-γ secretion ability of type I ILCs.

## Bulk RNA-seq depicts *Prdm1*-mediated functions in cNK cells

Bulk RNA sequencing of splenic cNK cells (CD3$^-$NK1.1$^+$NKp46$^+$) was conducted to uncover the molecular mechanisms by which *Prdm1* regulates NK cell anti-tumor immunity (*Figure 3A*). Differentially expressed genes (DEGs) between *Prdm1$^{+/+}$* and *Prdm1* cko mice were determined using a criterion of log$_2$ (fold change)>0.5 and p<0.05. 445 DEGs were identified out of 17434 protein-coding genes, which consisted of 223 upregulated genes and 222 downregulated genes (*Figure 3B*).

Gene Ontology (GO) analysis revealed the enrichment of glucuronate metabolism and lymphocyte differentiation in upregulated genes in *Prdm1* cko mice derived NK cells (*Figure 3C*), both of which were associated with cellular growth and development. In contrast, leukocyte-mediated cytotoxicity, immune receptor activity, and integrin binding were enriched in the genes which decreased their expression level in in *Prdm1* cko mice (*Figure 3C*). Gene Set Enrichment Analysis (GSEA) showed that NF-kappa B signaling pathway enriched in *Prdm1*-deficient cNK cells (*Figure 3D*), suggesting the potential targets by *Prdm1* to regulate NK cell function. Increased expression of multiple TFs such as *Junb*, *Batf3*, *Nfkb1*, *Tcf7*, and *Nr4a2* was observed in *Prdm1* cko cNK cells, suggesting they might be suppressed by *Prdm1* (*Figure 3E*). Downregulation of granzyme B (*Gzmb*), Perforin (*Prf1*) were observed in *Prdm1*-deficient NK cells (*Figure 3E*), implying decreased anti-tumor ability, which was consistent with increased melanoma metastasis in *Prdm1* cko mice (*Figure 2*, B and D). CXCR6 and CX3CR1 was considered to play an important role in promoting the egress of NK cells from bone marrow. Decreased expression of *Cxcr6* and *Cx3cr1* in *Prdm1* cko NK (*Figure 3E*) might be the reason for the increased quantity of NK cells in bone marrow and decreased number in peripheral tissues (*Figure 1G*). As a result of the reduced expression levels of *Cxcr6* and *Cx3cr1*, NK cells may not be able to egress from the bone marrow and accumulated therein. Consistent with decreased production of IFN-γ after stimulated by IL-12/IL-18 (*Figure 2F*), decreased expression of *Il18rap* and *Il12rb2* were observed in *Prdm1* cko cNK cells (*Figure 3E*), implying impaired response to cytokine stimulation.

To confirm the result of RNA-sequencing, the expression of fractalkine receptor (CX3CR1), granzyme B and perforin were analyzed by flow cytometry. The percentage of CX3CR1$^+$ cNK cells was significantly decreased in multiple tissues of *Prdm1* cko mice, while the proportion of CX3CR1$^+$ ILC1 was increased in the liver (*Figure 3*, F and G). Lower GZMB and PRF1 production was observed in *Prdm1*-deficient splenic cNK cells, liver cNK cells and ILC1s (*Figure 3*, H-K; *Figure 3—figure supplement 1*, A-I). Notably, the proportion of GZMB$^+$ and PRF1$^+$ cNK cells was decreased among almost all of the maturation stages of cNK cells (*Figure 3*, J and K). The relative mean fluorescent intensities (MFIs) of GZMB and PRF1 consistently show a reduction across all developmental stages in *Prdm* cko NK cells (*Figure 3—figure supplement 1*, H and I). Yet, no statistical difference of PRF1 was found within the CD11b$^-$CD27$^+$ and CD11b$^+$CD27$^+$ subsets, likely due to the relatively lower perforin levels in these populations (*Figure 3—figure supplement 1I*). These findings suggest that Prdm1 may directly influence cytotoxic molecule in NK cells, rather than impacting their anti-tumor abilities solely by affecting the maturation phenotype of Prdm1-deficient NK cells.

## scRNA-seq reveals distinct properties of two clusters of liver group I ILCs following *Prdm1* knockout

To further investigate the effect of *Prdm1* in liver cNK cells and ILC1s and the changes in the hepatic immune microenvironment caused by the deficiency of *Prdm1* in group 1 ILCs, single-cell RNA sequencing (scRNA-seq) was performed for liver CD45$^+$ cells (*Figure 4A*; *Figure 4—figure supplement 1A*). Initial quality control revealed high-quality of cell purity, library assembly, and sequencing (*Figure 4—figure supplement 1B*). 10,978 cells passed the quality criteria and were selected for further analysis (6,161 from *Prdm1$^{+/+}$* mice, 4,817 from *Prdm1* cko mice). Unsupervised clustering of all sequenced cells based on transcript signatures identified twelve distinct clusters (*Figure 4—figure supplement 1C*), including B cells, epithelial cells (ECs), CD4$^+$ T cells, CD8$^+$ T cells, NKT cells,

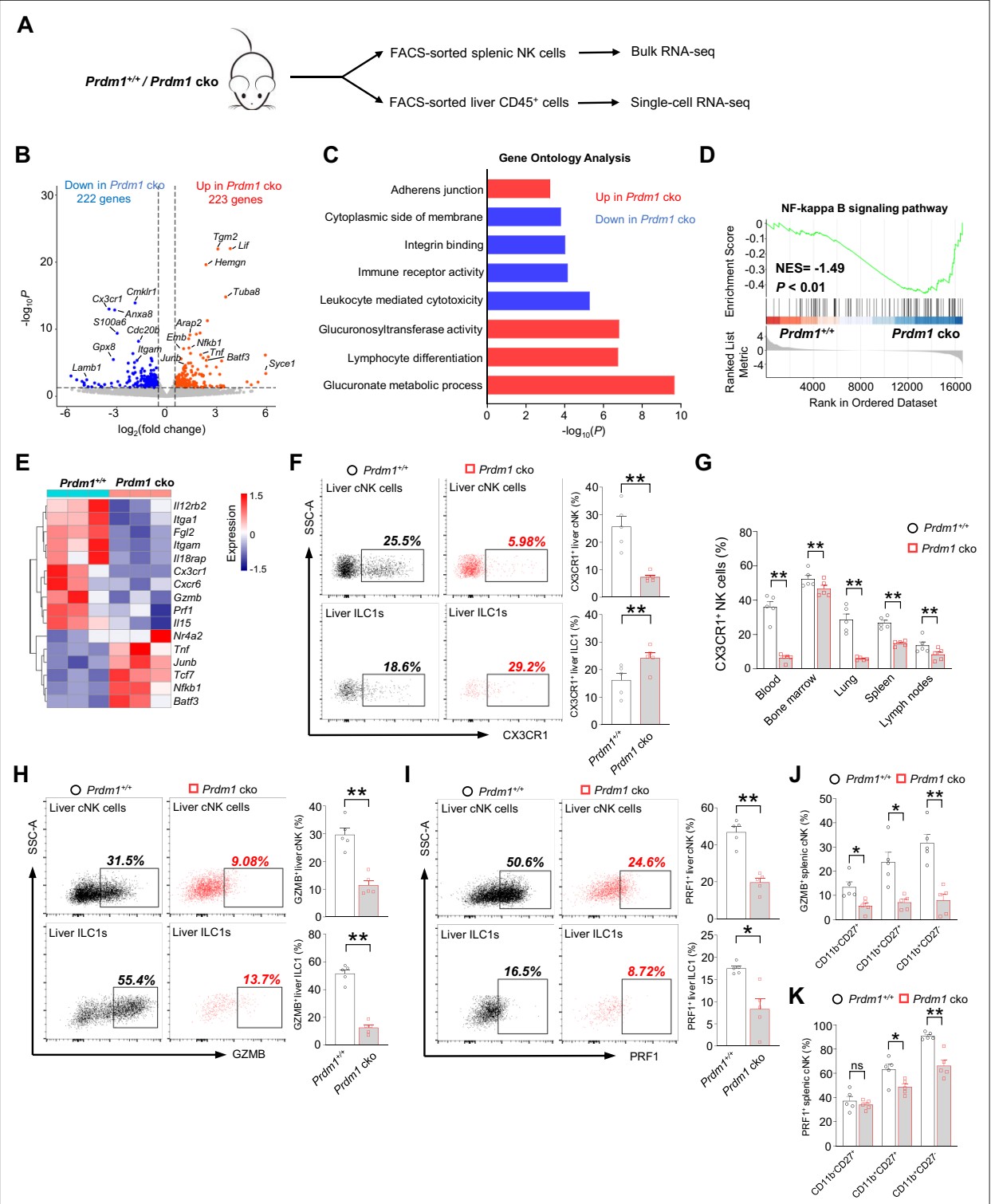

**Figure 3.** Bulk RNA-seq reveals Prdm1-mediated functions in splenic cNK cells. (**A**) Splenic cNK cells and liver CD45+ cells were sorted from *Prdm1+/+* and *Prdm1* cko mice using flow cytometry, and prepared for bulk RNA-seq and single-cell RNA-seq analysis. (**B**) Volcano plot of the bulk RNA-seq differentially expressed genes (DEGs) (log2|fold change|>0.5; p<0.05) in splenic cNK cells between *Prdm1+/+* and *Prdm1* cko mice. Upregulated and downregulated genes in *Prdm1* cko cells were highlighted in red and blue. (**C**) Enriched Gene Ontology (GO) terms of DEGs in *Prdm1* cko cells compared *Prdm1+/+* cells. The Enrichment gene set in upregulated (red) and downregulated (blue) genes were indicated in different colour. Bar length represents statistical significance. (**D**) Gene Set Enrichment Analysis (GSEA) showing the enrichment of NF-kappa B signaling pathway of DEGs in *Prdm1* cko cells compared *Prdm1+/+* cells. NES, normalized enrichment score. (**E**) Heatmap of selected genes from DEGs. Shown is z-score transformed

*Figure 3 continued on next page*

*Figure 3 continued*

expression of DEGs. (**F**) Representative flow cytometric plots (left) and cumulative data (right) of the percentage and absolute numbers of CX3CR1+ cells in liver cNK cells and ILC1s (n=5). (**G**) Quantification of CX3CR1+ cells in NK cells in blood, bone marrow, lung, liver, spleen, and lymph nodes (n=5). (**H and I**) Representative flow cytometric plots (left) and cumulative data (right) showing the proportion of GZMB+ (**H**) and PRF1+ (**J**) liver cNK cells and ILC1s from *Prdm1*+/+ and *Prdm1* cko mice (n=5). (**J and K**) Proportion of GZMB+ (**J**) and PRF1+ (**K**) splenic NK cells at different maturation stages was analyzed by flow cytometry (n=5). Data are presented as the mean ± SEM and were analyzed by two-tailed, paired t-test. Differences were evaluated between littermates. Each circle and square on graphs represents an individual mouse; *P*, p-value; *, p<0.05; **, p<0.01, ns, not significant.

The online version of this article includes the following figure supplement(s) for figure 3:

**Figure supplement 1.** *Prdm1* deficiency impairs production of Granzyme B and Perforin in group 1 ILC.

ILC1s, cNK cells, dendritic cells (DCs), monocyte-derived macrophages (MDMs), Monocytes, Kupffer cells (KCs), Neutrophils, and a small number of undefined cells (*Figure 4A*). In *Prdm1* cko mice, an increased proportion of MDMs, CD4+ T cells, CD8+ T cells, Monocytes, and DCs was observed, alongside a decreased proportion of ECs, cNK cells, ILC1s, and NKT cells (*Figure 4—figure supplement 1D*). Liver cNK cells and ILC1s were identified and discriminated based on the expression of surface markers and distinctive TFs (*Figure 4—figure supplement 1*, E and F). Compared with other clusters, cNK cells and ILC1s highly expressed *Ncr1* and *Klrb1c* (NK1.1; *Figure 4—figure supplement 1E*). cNK cells expressed high levels of *Itga2* (CD49b) and Eomes, while ILC1s had high levels expression of *Itga1* (CD49a) and *Tnfsf10* (*Figure 4—figure supplement 1*, F and G). Consistent with our flow cytometry data, both cNK cells and ILC1s have significant reduced proportion in *Prdm1* cko mouse (*Figure 4—figure supplement 1H*). In group 1 ILCs from *Prdm1* cko mice, there was an increase in the proportion of cNK cells accompanied by a decrease in ILC1s (*Figure 4*, B and C).

To better understand the specific function of *Prdm1* in liver cNK cells and ILC1s, the two subpopulations of liver group 1 ILCs were further analyzed separately using unsupervised clustering and visualized by Uniform Manifold Approximation and Projection (UMAP; *Figure 4*, D and E). Based on the cluster specific gene expression signature (*Figure 4—figure supplement 2A*), the subpopulation of liver cNK cells were referred as '*Prf1* high', '*Junb* high', and '*Cxcr3* high' cNK cells (*Figure 4D*), with different distribution in *Prdm1* cko and *Prdm1*+/+ genotype mice (*Figure 4F*; *Figure 4—figure supplement 2B*).

The *Prf1* high cNK cell cluster was defined by high expression of cytolysis-related genes, including *Ncr1*, *Gzma*, *Gzmb*, *Prf1*, and *Fgl2* (*Figure 4G*; *Figure 4—figure supplement 2*, C and D), indicating the strong target-killing ability of this cluster. Although this cluster is present in both *Prdm1* cko and *Prdm1*+/+ mice, there is a significant reduction in *Prdm1* cko mice (*Figure 4F*), indicating the importance of Prdm1 in maintaining this group of cells. GO analysis further revealed the enrichment signatures of cytolysis, response to virus, and lymphocyte-mediated immunity in the genes upregulated in *Prf1* high cNK cell cluster, further confirming the cytotoxic effects of this cluster (*Figure 4H*). These data underscore the crucial role of *Prdm1* in maintaining NK cells with immune effector functions.

The *Junb* high liver cNK cell cluster distinguished themselves by higher expression of *Junb* compared to other clusters (*Figure 4G*; *Figure 4—figure supplement 2*, C and D). The predominant majority (92.98%) of *Junb* high liver cNK cells are derived from *Prdm1* cko mice, with less than ten percent (7.02%) originating from *Prdm1*+/+ mice (*Figure 4F*). Many signal transduction elements, gene expression regulator, and transcriptional factors, such as *Nfkbia*, *Tnfaip3*, *Nr4a1/2/3*, *Batf3*, *Fos*, *Fosb*, *Tcf7*, and *Kit* were upregulated in the *Junb* high liver cNK cells (*Figure 4G*; *Figure 4—figure supplement 2D*). The expression of cytotoxicity related genes, such as *Gzmb* and *Prf1*, in *Junb* high cluster was lower than other cNK cell clusters (*Figure 4G*). GO analysis showed that genes upregulated in *Junb* high liver cNK cells enriched in cell differentiation, cell activation, and transcriptional regulation (*Figure 4H*). GSEA indicates that the NF-kappa B, IL-17, MAPK, and TNF signaling pathways were upregulated in this clusters (*Figure 4*, I-L). GSEA also showed that mitochondrial related pathways, such as mitochondrial protein, oxidative phosphorylation, and respiratory electron transport chain were suppressed in *Junb* high cNK cell cluster (*Figure 4—figure supplement 2*, E-G). Increased proportion of *Junb* high cluster in *Prdm1* deficient cNK cells suggested impaired anti-tumor activity, which was consistent with more melanoma metastasis in *Prdm1* cko mice and lower expression of cytotoxicity-related genes in splenic cNK cells based on bulk RNA-sequencing.

The *Cxcr3* high cNK cell cluster was characterized with high expression of *Cxcr3*, *Ccr2*, and some genes encoding ribosomal subunits such as *Rps7* (*Figure 4—figure supplement 2A*). Expression of

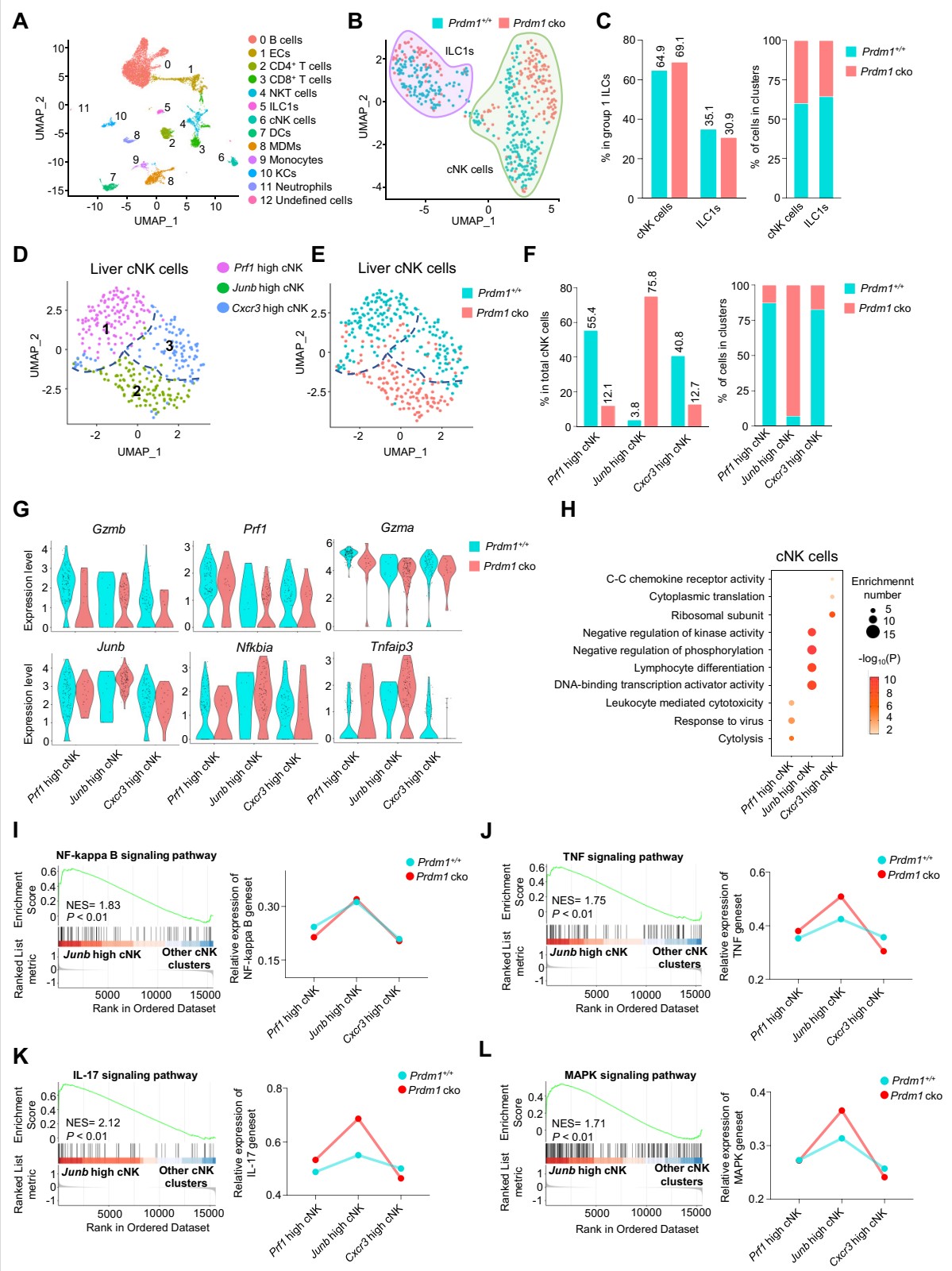

**Figure 4.** Different properties of cNK clusters following *Prdm1* knockout. (**A**) Uniform manifold approximation and projection (UMAP) visualization of liver CD45+ cells from *Prdm1+/+* and *Prdm1* cko mice. Twelve clusters were defined and indicated by distinct colours. Each dot represents a single cell. (**B**) UMAP visualization of liver cNK and ILC1 clusters. Cells were colored by genotypes (*Prdm1+/+*-blue; *Prdm1* cko-red). (**C**) Percentages of cNK cells and ILC1s in total group 1 ILCs (left), and their distribution in each cluster (right). (**D and E**) UMAP visualization of three different liver cNK clusters

*Figure 4 continued on next page*

*Figure 4 continued*

from two mouse strains. (**F**) Proportions of cNK cells among total cNK cells (left; 211 cells in *Prdm1*[+/+], and 141 cells in *Prdm1* cko) and within clusters (right). (**G**) Violin plots showing the normalized expression of select genes in different cNK clusters. (**H**) Enriched GO term of marker genes in three cNK clusters. Dot size represents enriched gene number, and color intensity represents significance. (**I–L**) GSEA plots (left) depicting the enrichment of NF-kappa B (**I**), TNF (**J**), IL-17 (**K**), and MAPK (**L**) signaling pathway in *Junb* high cNK cluster compared with clusters of *Prf1* high and *Cxcr3* high cNK cells. Right panel showed dynamic relative expression of the given gene sets from cluster1 to cluster3 between *Prdm1*[+/+] and *Prdm1* cko. Dots represent the average expression of given gene set in each cell, which was calculated through the sum of normalized expression of each individual gene within the designated gene set in every single cell. NES, normalized enrichment score.

The online version of this article includes the following figure supplement(s) for figure 4:

**Figure supplement 1.** scRNA-seq identified subsets of liver CD45[+] cell from *Prdm1*[+/+] and *Prdm1* cko mice.

**Figure supplement 2.** Cluster-specific markers of liver cNK cell clusters.

tissue-resident markers *Cd69* was also highly expressed in this clusters (*Figure 4—figure supplement 2D*). The enrichment of chemokine receptors in genes upregulated in the *Cxcr3* high cluster implied a greater likelihood of this cluster being tissue-resident compared with other cNK cell clusters (*Figure 4H*). To further confirm tissue-resident properties of this clusters, we calculated the module score based on top30 DEGs in ILC1 versus cNK clusters, including *Cxcr6*, *Itga1*, *Cd160*, *Cd226*, etc. *Cxcr3* high cNK clusters have the highest score among all cNK clusters (*Figure 4—figure supplement 2H*), indicating the similarity with liver ILC1s. In the tumor microenvironment, reports indicated that NK cells could transform into ILC1s (*Gao et al., 2017*). If this conversion of cNK cells into ILC1s also occurred under normal physiological conditions then *Cxcr3* high cNK cell cluster might be the most susceptible to such transformation.

The significant enrichment of ribosomal subunits and cytoplasmic translation in *Cxcr3* high cluster (*Figure 4H*) implied their distinct and active metabolic profile and the capability to mount immune responses. The remarkably decreased proportion of *Cxcr3* high cNK cell cluster in *Prdm1* cko mice (*Figure 4F*) emphasized the critical role of *Prdm1* in maintaining this cluster of liver cNK cells, consistent with the flow cytometry result that showed an increase in the number of NK cells in the bone marrow and a decrease in NK cells in peripheral tissues (*Figure 1G*). This is consistent with flow cytometry data, which showed a decrease in the proportion of CX3CR1[+] cNK cells across all tested organs following Prdm1 knockout. However, it is noteworthy that, in contrast to the trend in cNK cells, CX3CR1 expression was increased in liver ILC1s after Prdm1 knockout (*Figure 3*, F and G). This not only substantiates the involvement of Prdm1 in managing NK cell migration but also underscores its distinctive regulatory impacts on the chemokine receptor expressions within NK cells and ILC1 populations. Bulk RNA sequencing data also found that the expression of *Cx3cr1* and *Cxcr6* decreased in *Prdm1* cko cNK cells (*Figure 3E*). These findings supported the hypothesis that *Prdm1*, through regulating chemokine receptor expression levels, influenced the distribution of NK cells in the bone marrow and peripheral tissues, particularly within the liver tissue.

Three clusters of ILC1s were identified from liver ILC1s (*Figure 5A*), comprising 'Il7r high', 'Klra high', and 'Gzmb high' ILC1s. *Prdm1*[+/+] and *Prdm1* cko ILC1s seemed to co-cluster largely and have minor difference within the proportion of clusters (*Figure 5*, B and C; *Figure 5—figure supplement 1B*). The first two clusters of ILC1s were characterized by higher expression of *Il7r* (CD127) and *Klra5* separately, while the *Gzmb* high ILC1 cluster was identified by elevated expression of both *Gzma* and *Gzmb* (*Figure 5D*; *Figure 5—figure supplement 1*, A, C and D). Additionally, both *Gzma* and *Gzmb* expression were downregulated, and *Junb* was upregulated, in *Prdm1* cko mice derived cNK cells and ILC1s compared to those from *Prdm1*[+/+] mice (*Figure 5E*). The *Il7r* high ILC1s cluster was distinguishable from other ILC1 clusters by its high expression of *Il7r*, *Il18r1* (IL18RA), and *Ifng* (IFN-γ) (*Figure 5B*; *Figure 5—figure supplement 1C*). The high expression of *Il18r1* and *Ifng* in *Il7r* high ILC1s indicated this cluster of cells was highly responsive to IL-18 (*Figure 5—figure supplement 1D*). GSEA and GO analysis showed that IL-17, NF-kappa B, TNF, MAPK signaling pathway and T cell differentiation were activated in the *Il7r* high ILC1 cluster (*Figure 5*, F-H; *Figure 5—figure supplement 1*, E-G). Module scores, calculated based on the expression of feature genes within the *Junb* high cNK cell cluster, revealed a comparable *Junb* high signature expression pattern within the *Il7r* high ILC1 cluster (*Figure 5I*). Several ILC3 signature genes, such as *Rora*, *Tmem176a*, and *Tmem176b* (*Robinette et al., 2015*), highly expressed in this cluster (*Figure 5—figure supplement 1D*). Considering the close relationship between IL-17-mediated immunity response and ILC3 (*Spits et al., 2013*; *Klose et al.,*

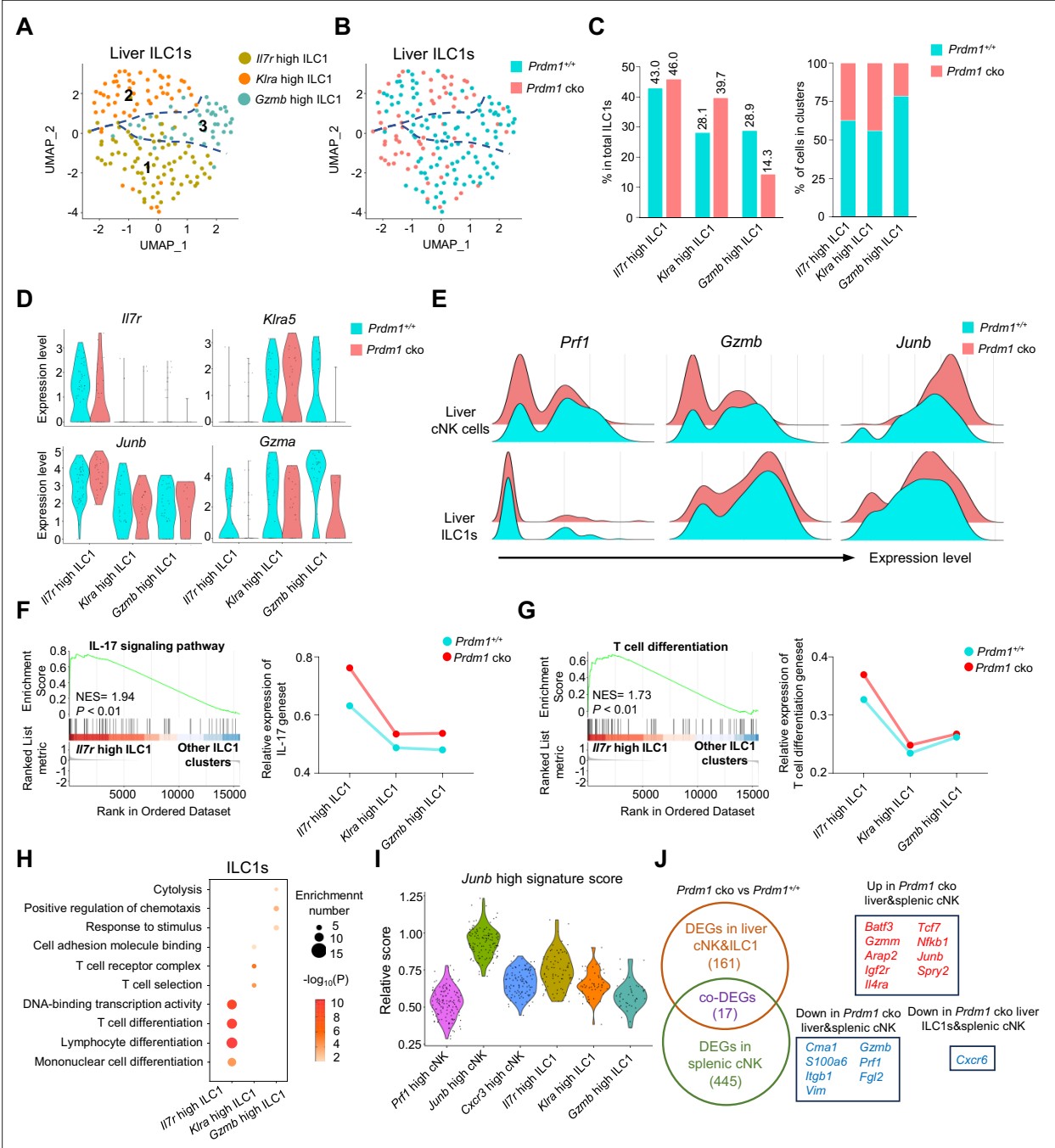

**Figure 5.** Different properties of ILC1 clusters following *Prdm1* knockout. (**A and B**) UMAP visualization of three different liver ILC1 clusters from two mouse strains. (**C**) Proportions of ILC1s among total ILC1s in different genotypes (left; 114 cells in *Prdm1*⁺/⁺, and 63 cells in *Prdm1* cko) and within each cluster (right). (**D**) Violin plots showing the normalized expression of select genes in different ILC1 clusters. (**E**) Ridge plots showing the normalized expression of *Gzmb*, *Prf1*, and *Junb* in cNK and ILC1 clusters between *Prdm1*⁺/⁺ and *Prdm1* cko cells. (**F and G**) GSEA plots (left) depicting the enrichment of IL-17 signaling pathway (**F**) and T cell differentiation (**G**) in *Il7r* high ILC1 cluster compared with clusters of *Klra* high and *Gzmb* high ILC1s. Right panel showed dynamic relative expression of the given gene sets from cluster1 to cluster3 between *Prdm1*⁺/⁺ and *Prdm1* cko. Dots represent the average expression of given gene set in each cell, which was calculated through the sum of normalized expression of each individual gene within the designated gene set in every single cell. NES, normalized enrichment score. (**H**) Enriched GO term of marker genes in three ILC1 clusters. Dot size represents enriched gene number, and color intensity represents significance. (**I**) Violin plot showing the *Junb* high signature score for cNK cell and ILC1 clusters, calculated using the signature genes of *Junb* high cNK cluster. (**J**) Venn diagram showing overlapping and unique DEGs in comparisons within liver cNK cells, ILC1s and splenic cNK cells between *Prdm1*⁺/⁺ and *Prdm1* cko(left), and 17 overlapped DEGs were shown at the right panel.

The online version of this article includes the following figure supplement(s) for figure 5:

*Figure 5 continued on next page*

Figure 5 continued

**Figure supplement 1.** Cluster-specific markers of liver ILC1 clusters.

**Figure supplement 2.** Similar and unique gene expression patterns in group 1 ILCs regulated by *Prdm1* and *Hobit*.

*2013*), it is plausible that *Il7r* high ILC1 cluster may be attributed, at least in part, to potential plasticity between ILC1 and ILC3 subsets.

The second liver ILC1 cluster, characterized by high expression of *Klra5* (Ly49E) and *Klra7* (Ly49G), was designated as the *Klra* high ILC1 cluster (*Figure 5D*; *Figure 5—figure supplement 1D*). Notably, there was an elevated proportion of *Klra* high ILC1s in *Prdm1* cko ILC1s (39.7%) compared to *Prdm1*$^{+/+}$ ILC1 s (28.1%; *Figure 5C*). Liver Ly49E$^+$ ILC1s have been identified as possessing greater cytotoxic potential and a more robust viral response compared to liver Ly49E$^-$ ILC1s (*Chen et al., 2022*). The *Klra* high cluster exhibited notably high expression of *Ccl5* (*Figure 5—figure supplement 1D*). Previous research has underscored the pivotal role of CCL5, produced by both cNK cells and ILC1s, in facilitating the accumulation of DCs within the tumor microenvironment, thereby impeding tumor immune evasion, as highlighted in studies (*Böttcher et al., 2018*; *Kirchhammer et al., 2022*). The expression of *Ccl5* was reduced in the *Klra* high cluster of *Prdm1* cko ILC1s compared to *Prdm1*$^{+/+}$ ILC1 s (*Figure 5—figure supplement 1D*), which could potentially have a detrimental impact on the ability of *Klra* high ILC1s to develop a connection between innate and adaptive immune responses.

The *Gzmb* high ILC1 cluster was identified according to the high expression of *Gzma*, *Gzmb*, and *Fgl2* (*Figure 5—figure supplement 1D*) compared to other ILC1 clusters. GO analysis also revealed the enrichment in cytolysis and stimulus-response capacity (*Figure 5H*) of the *Gzmb* high ILC1 cluster. Consistent with the *Prf1* high cNK cell cluster, the proportion of the *Gzmb* high cluster among liver ILC1s exhibited a considerable reduction in *Prdm1* cko mice compared to *Prdm1*$^{+/+}$ mice, and the expression of *Gzma* also downregulated in *Prdm1* cko mice (*Figure 5B*). Previous reports showed that GzmA$^+$ ILC1 constituted the main population of liver ILC1s at birth, with the potential target-killing ability (*Di Censo et al., 2021*; *Friedrich et al., 2021*). Within cNK cells, *Il12rb2*, *Il18r1* and *Il18rap* was highly expressed in *Prf1* high and *Cxcr3* high cNK clusters (*Figure 4—figure supplement 2I*), indicating the IL-18 receptor expression correlated with the NK cell maturation. While in ILC1, these receptors mostly expressed on *Il7r* high and *Gzmb* high ILC1 clusters (*Figure 5—figure supplement 1D*). Significant decreased of *Il18r1* expression in *Prdm1* cko cNK cells and ILC1s may associated with the impaired ability to produce IFN-γ.

To investigate the universal transcriptional program between group 1 ILCs across liver and spleen, we have explored DEGs in *Prdm1* cko liver cNK cells and ILC1s using our scRNA-seq data. Compared with liver ILC1s, more DEGs was observed in liver cNK cells, including *Junb*, *Kit*, *Tcf7*, *Gzmb*, *Prf1*, etc. (*Figure 5—figure supplement 2*, A and B). Through the integration of the bulk RNA-seq and scRNA-seq data, we identified 17 DEGs that are regulated by Prdm1 among liver cNK cells, splenic cNK cells, and liver ILC1s. *Batf3*, *Junb*, *Tcf7*, and *Nfkb1* was upregulated, whereas *Gzmb*, *Prf1*, and *Fgl2* downregulated, in both *Prdm1* cko liver and splenic cNK cells (*Figure 5J*). Cxcr6 was downregulated in liver ILC1s and splenic cNK cells in *Prdm1* cko mice (*Figure 5J*). Previous research found that spleen NK cells could be divided into three distinct groups based on their expression levels of CD27, CD62L, CD49a, and CD49b (*Flommersfeld et al., 2021*). CD27$^+$CD62L$^-$ NK cells have remarkable high expression of Batf3, while it was only barely expressed in CD27$^+$CD62L$^+$ and CD27$^-$CD62L$^+$ NK cells (*Flommersfeld et al., 2021*). Based on the sequencing data published by Flommersfeld et al. (GSE180978), a notable negative correlation was observed between the expression levels of Prdm1 and Batf3 (*Figure 5—figure supplement 2I*). On top of that, our findings unveiled the negative regulatory influence of Prdm1 on Batf3 within both spleen and liver NK cells. This discovery highlights a potential upstream mechanism that may influence the hemostasis of the spleen NK cell subpopulations through Batf3.

We also compared the gene regulation patterns between *Prdm1* and *Hobit* (homologue of Blimp1) based on two published scRNA-seq data (*Friedrich et al., 2021*; *Yomogida et al., 2021*). Following the knockout of Hobit, the DEGs were primarily identified within ILC1s. Conversely, after the knockout of Prdm1, a greater number of DEGs were observed in cNK cells. This indicates that Prdm1 likely possesses a broader range of target genes within cNK cells, whereas Hobit appears to have a more pronounced impact on gene expression within ILC1s (*Figure 5—figure supplement 2*, C-F). There are some overlaps between the downstream transcriptional profile of *Prdm1* and *Hobit* in liver cNK cells

and ILC1s (*Figure 5—figure supplement 2*, G and H). Specifically, genes such as *Junb*, *Fosb*, *Tcf7*, *Kit*, *Gzmb*, *Prf1*, and *Cxcr6* were simultaneously upregulated or downregulated in both *Prdm1* cko and *Hobit*[KO] liver cNK cells or ILC1s, indicating the similar regulatory networks of Prdm1 and Hobit.

## *Prdm1* facilitates the intercellular communication between liver group 1 ILCs and macrophages

The reciprocal crosstalk between group 1 ILCs and macrophages plays a critical role in maintaining liver immune homeostasis and anti-cancer immune surveillance (*Tu et al., 2008*; *Park et al., 2023*). The scRNA sequencing analysis identified two well-established subpopulations of liver macrophages: the resident Kupffer Cells (KCs) and the Monocyte-Derived Macrophages (MDMs) (*Figure 6*, A-C; *Figure 6—figure supplement 1A*). When comparing the total proportion of macrophages within the immune cell population of the liver between *Prdm1*[+/+] and *Prdm1* cko mice, there is an increase in *Prdm1* cko mice (*Figure 6C*). To confirm these findings, we utilized flow cytometry to define macrophages, including both KCs and MDMs, gating by CD45[+]Ly6G[-]F4/80[+]CD11b[+] (*Figure 6D*). Our analysis showed that, following the deletion of *Prdm1* in Group 1 ILCs, there is a significant increase in both the proportion and number of macrophages in the liver (*Figure 6D*).

According to the transcriptional profile, liver macrophages were further clustered and labeled as '*Ly6c2* high'; '*Cxcl2* high'; '*Ear2* high' MDMs, and '*Mrc1* high'; '*C1q* high' KCs (*Figure 6A*, *Figure 6—figure supplement 1*, A-E). Increased proportion of MDMs and KCs was observed in *Prdm1* cko cells, which was consistent with flow cytometry data (*Figure 6—figure supplement 1B*, and *Figure 6D*). Within MDMs clusters, *Ly6c2* high MDMs mainly compose of *Prdm1*[+/+] cells, while *Prdm1* cko cells concentrated in *Cxcl2* high cluster (*Figure 6C*). The scRNA-seq data revealed that following Prdm1 knockout in NKp46[+] cells, there was a decrease in the proportion of KCs within the macrophage population, while the proportion of MDMs increased (*Figure 6D*). CX3CR1, a chemokine receptor, is extensively utilized to distinguish KCs and MDMs within macrophages. Cells expressing CX3CR1 are classified as MDMs, whereas those without CX3CR1 expression are categorized as KCs (*Heymann et al., 2015*). Using flow cytometry and assessing CX3CR1 expression, we determined the ratios of KCs and MDMs. In contrast to the scRNA-seq findings, flow cytometry indicates that following *Prdm1* knockout in group 1 ILCs, there is a minor increase in the proportion of KCs within the total liver macrophages, and a decrease in the proportion of MDMs (*Figure 6D*; *Figure 6—figure supplement 1B*). This discrepancy might stem from the different bases of classification: scRNA-seq defines KCs based on gene expression profiles, whereas flow cytometry differentiates between KCs and MDMs using the single surface marker, CX3CR1. Analysis of the macrophage subsets identified by scRNA-seq reveals that, while MDM clusters generally show high CX3CR1 expression, there exists a subset within MDMs, labeled *Mrc1* high, that also exhibits high levels of CX3CR1 (*Figure 6—figure supplement 1C*). Consequently, if flow cytometry solely employs CX3CR1 for differentiating between KCs and MDMs, it could result in disparities when compared to scRNA-seq data. Both KCs and MDMs has significantly increased in *Prdm1* cko mice, which was consist with the scRNA-seq data (*Figure 6—figure supplement 1*, B and F). Despite the decrease in the proportion of *Ly6c2* high MDMs in *Prdm1* cko mice, the expression levels of *Ly6c2* exhibited minimal variation between *Prdm1*[+/+] and *Prdm1* cko mice (*Figure 6—figure supplement 1D*). Intriguingly, within certain cellular subsets, notably the *Ear2* high cluster, the *Ly6c2* expression levels in *Prdm1* cko mice were found to be higher than those in *Prdm1*[+/+] mice. Additionally, we employed flow cytometry to examine Ly6C expression within the macrophages. Similar with the scRNA-seq findings, there were no notable differences in Ly6C expression levels between *Prdm1*[+/+] and *Prdm1* cko mice (*Figure 6E*; *Figure 6—figure supplement 1G*).

High-resolution interactions among liver cNK cells, ILC1s, and macrophages were established and compared between *Prdm1*[+/+] and *Prdm1* cko mice using the CellChat program (*Jin et al., 2021*). Interactions between ILC1s and total macrophages were higher than that between cNK cells and macrophages (*Figure 6—figure supplement 2*, A, C, E, and G). Cross-talk between liver group 1 ILCs with macrophages enriched in macrophage migration inhibitory factor (MIF), MHC-I, CXC chemokine ligand (CXCL), Thy-1 cell surface antigen (THY1), and C-type lectin (CLEC) pathways (*Figure 6—figure supplement 2*, B, D, F, and H). Although the quantity of macrophages significantly increases in *Prdm1* cko mice, there is a significant decrease in the interaction number and interaction strength between liver group 1 ILCs and macrophages (~1.5 fold) in *Prdm1* cko mice (*Figure 6*, F and G). The reduction of interaction mostly occurred in the cross-talk of ILC1-MDM and ILC1-KC, whereas no difference was

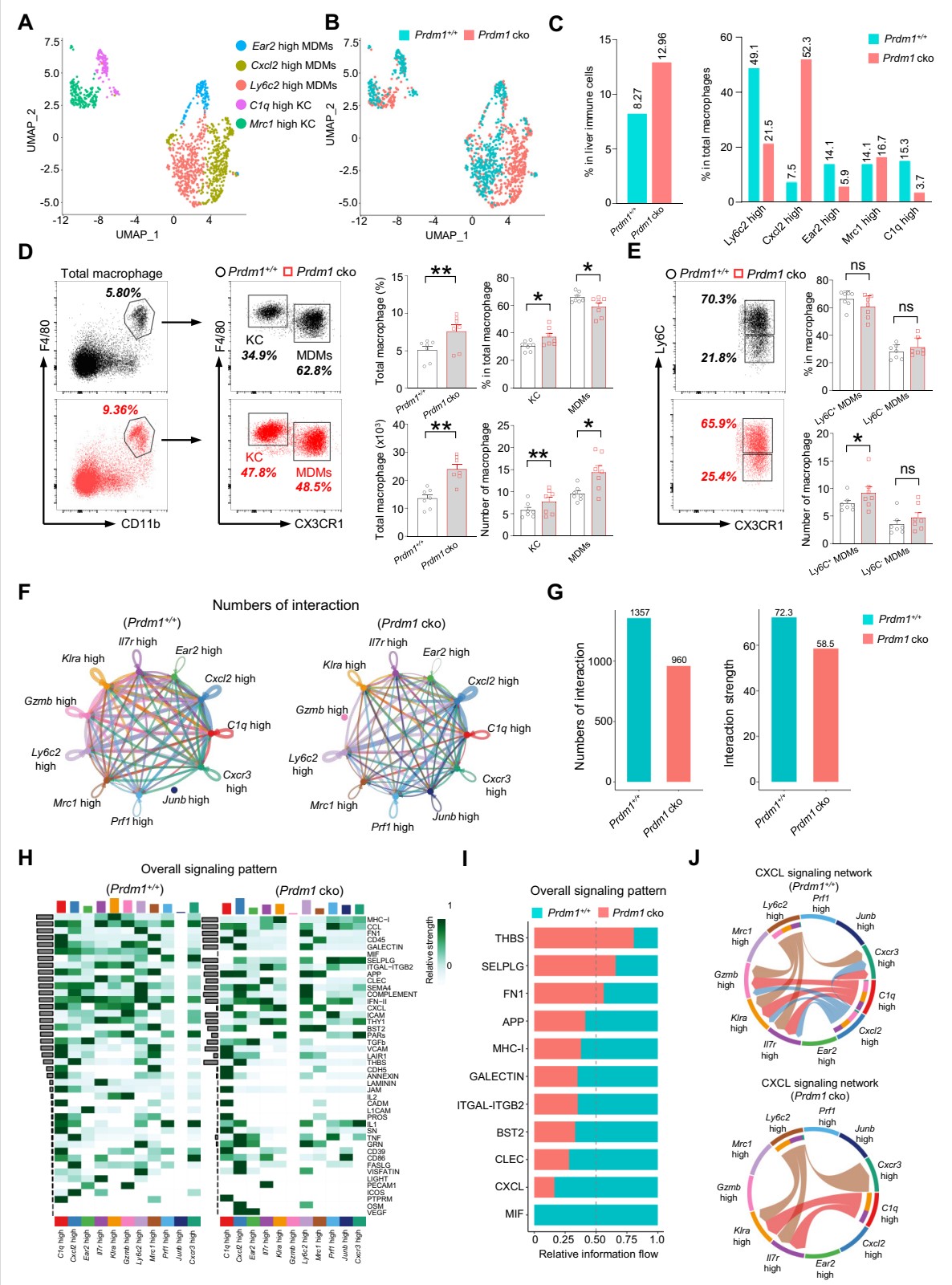

**Figure 6.** *Prdm1* facilitates the intercellular communication between liver group 1 ILCs and macrophages. (**A and B**) UMAP visualization of monocyte-derived macrophages (MDMs) and Kupffer cells (KCs) cluster (**A**) between *Prdm1*^+/+^ and *Prdm1* cko (**B**). (**C**) Proportions of total macrophages in liver immune cells (left), and proportions of MDMs and KCs among total macrophages in different genotypes (510 cells in *Prdm1*^+/+^, and 624 cells in *Prdm1* cko). (**D**) Representative flow cytometric plots (left) and cumulative data (right) of the percentage and absolute numbers of liver total macrophages

*Figure 6 continued on next page*

*Figure 6 continued*

(CD45+Ly6G-CD11b+F4/80+), MDMs (CX3CR1-), and KCs (CX3CR1+) between *Prdm1+/+* and *Prdm1* cko (n=7). (**E**) Representative flow cytometric plots (left) and cumulative data (right) of the percentage and absolute numbers of Ly6C- and Ly6C+ cells in MDMs. (**F and G**) Circle plots (**F**) and summary data (**G**) illustrating the interaction numbers and strength of significant enriched ligand–receptor pairs among cluster of liver cNK cells, ILC1s, and macrophages from *Prdm1+/+* (left) and *Prdm1* cko (right) cells. The thickness of the line indicates the number of enrich pairs, and the arrow reflects the direction of the interaction. (**H**) Heatmap of overall signaling pattern recognized from ligand-receptor pairs, which contained the sum of signaling from the sender and target cells. (**I**) Bar graphs showing the information flow in selected active signaling patterns between *Prdm1+/+* and *Prdm1* cko cells. Relative information flow was calculated as the sum of the communication probability in given signaling patterns. (**J**) Chord plot of the CXCL signaling interaction network among cluster of liver cNK cells, ILC1s, and macrophages between *Prdm1+/+* and *Prdm1* cko cells. Data are presented as the mean ± SEM and were analyzed by two-tailed, paired t-test. Differences were evaluated between littermates. Each circle and square on graphs represents an individual mouse; *P*, p-value; *, p<0.05; **, p<0.01, ns, not significant.

The online version of this article includes the following figure supplement(s) for figure 6:

**Figure supplement 1.** Identification of distinct macrophage clusters.

**Figure supplement 2.** Cell-Cell communication between liver cNK, ILC1s, MDMs, and KCs.

**Figure supplement 3.** Ligand-receptor interaction between type I ILCs and macrophages.

observed in cNK-MDM and cNK-KC interaction (*Figure 6—figure supplement 2*, A-H). A reduction in the interaction of ligand-receptor, such as Mif-CD74, Cxcl16-Cxcr6, and Cxcl10-Cxcr3 was observed in *Prdm1* cko mice compared to *Prdm1+/+* mice (*Figure 6—figure supplement 3*). Compared to *Prdm1+/+* mice, the information flow of CXCL and MIF pathways significantly decreased in *Prdm1* cko mice (*Figure 6*, H and I; *Figure 6—figure supplement 2*, B, D, F, and H). These pathways play a crucial role in facilitating macrophage migration. The CXCL signaling was sent from *Ly6c2* high *Cxcl2* high MDMs and *C1q* high KC, targeting all ILC1 clusters and *Cxcr3* high cNK cell clusters (*Figure 6J*). Of note, although the population of *Cxcl2* high macrophage primarily comprised cells from *Prdm1* cko mice, the interaction within the CXCL pathway between macrophages and group 1 ILCs was obviously less than *Prdm1+/+* sample (*Figure 6J*). These changes could be linked to a decreased population of ILC1s and *Cxcr3* high cNK cell cluster in *Prdm1* cko mice, implying that the homeostasis of *Cxcl2* high macrophages required sufficient signals from cNK cells and ILC1s. The impaired CXCL-CXCR interactions might subsequently lead to reduced recruitment and activation of group 1 ILCs and macrophages within the tumor microenvironment.

### *Prdm1* safeguards group 1 ILCs from exhaustion-like phenotypes in the tumor microenvironment

The suppression of mitochondrial related pathways in *Junb* high cNK cell cluster, along with a significant increase of this cNK cell cluster in *Prdm1* cko mice, encouraged us to explore mitochondrial function through flow cytometry. MitoTracker, MitoSOX, and Tetramethylrhodamine methyl ester (TMRM) were used to assess the mitochondrial mass, superoxide production, and mitochondrial membrane potential. A substantial decrease in MFIs of MitoTracker was observed in *Prdm1* cko splenic cNK cells, liver cNK cells, and liver ILC1s when compared to their *Prdm1+/+* counterparts (*Figure 7A*). This observation aligns with the enrichment of downregulated genes from *Prdm1*-deficient sample in mitochondrial related pathway, as revealed by RNA sequencing data (*Figure 4—figure supplement 2*, D-F). There was no significant difference in MitoSOX and TMRM between *Prdm1* cko and *Prdm1+/+* mice (*Figure 7*, B and C), which suggested that the ATP synthesize capacity was minimally affected by *Prdm1*.

IFN-γ is a critical cytokine for NK cells mediated cancer surveillance (*Takeda et al., 2011*; *Lin et al., 2021*) and impaired production of IFN-γ was considered as a key hallmark of exhausted NK cells (*Roe, 2022*; *Zhang et al., 2018*). To evaluate the IFN-γ secreting capacity of liver cNK cells and ILC1s in tumor microenvironment, B16F10 tumor cells were inoculated to the liver via splenic injection and the IFN-γ levels in response to stimulation of IL-12 and/or IL-18 were assessed by flow cytometry (*Figure 7—figure supplement 1A*). The proportion changes of cNK cells and ILC1s in *Prdm1* cko mice was similar with the no tumor-burden condition, while the number of both cNK cells and ILC1s have significant decreased in tumor-bearing liver (*Figure 7D*). Compared with *Prdm1+/+* mice, significant deceased of IFN-γ+ cells were observed in *Prdm1* cko mice liver cNK cells and ILC1s under the combinate stimulation of IL-12/IL-18 (*Figure 7E*; *Figure 7—figure supplement 1B*), which was more remarkable in liver ILC1s. Similar trends were observed when IL-12 or IL-18 was used alone, although

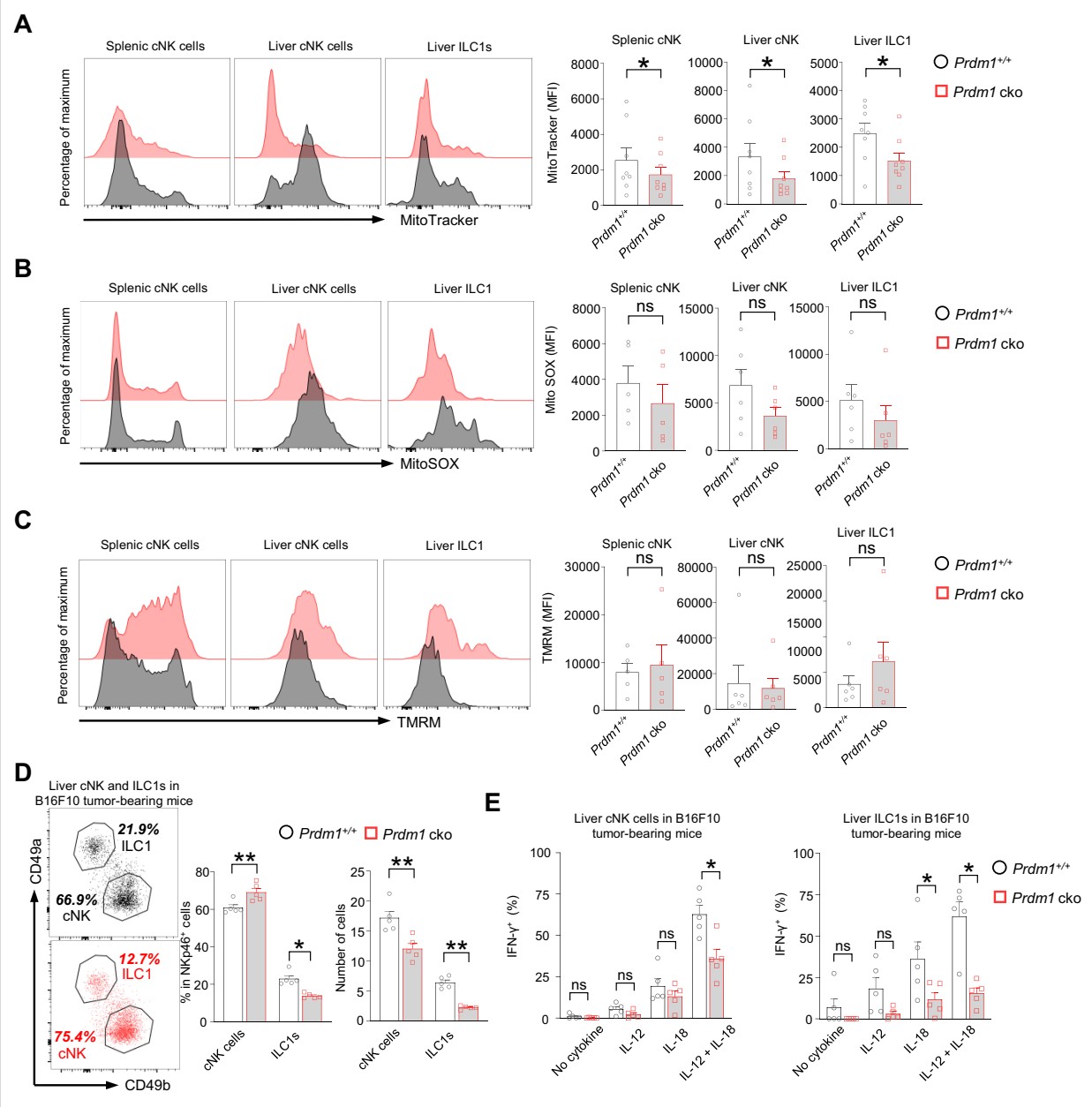

**Figure 7.** *Prdm1* safeguards group 1 ILCs from exhaustion-like phenotypes in the tumor microenvironment. (**A–C**) The Mitochondrial mass (MitoTracker Green staining; n=8) (**A**), Mitochondrial ROS (MitoSOX staining; n=5) (**B**), and Mitochondrial membrane potential (TMRM staining; n=5) (**C**) of splenic cNK cells, liver cNK cells and ILC1s were analyzed by flow cytometry. Representative flow cytometric plots (left) and cumulative data (right) showing the relative mean fluorescence intensities (MFIs) of each group. (**D**) Representative flow cytometric plots (left) and cumulative data (right) showing the percentage and absolute number of liver cNK cells and ILC1s in *Prdm1*[+/+] and *Prdm1* cko tumor-bearing mice at day 14 after inoculation with B16F10 melanoma cells via intrasplenic injection(n=5). (**E**) Percentages of IFN-γ[+] liver cNK cells and ILC1s from *Prdm1*[+/+] and *Prdm1* cko tumor-bearing mice (n=5). Data are presented as the mean ± SEM and were analyzed by two-tailed, paired t-test. Differences were evaluated between littermates. Each circle and square on graphs represents an individual mouse; *P*, p-value; *, p<0.05; **, p<0.01, ns, not significant.

The online version of this article includes the following figure supplement(s) for figure 7:

**Figure supplement 1.** *Prdm1* maintains the IFN-γ production of liver type 1 ILCs in tumor microenvironment.

only liver ILC1s showed a significant decrease in response to IL-18 stimulation (*Figure 7E*). These findings were consistent with the heavy tumor burden observed in *Prdm1* cko mice.

## Discussion

Prdm1 is a pivotal transcription factor that has attracted substantial research interest due to its role in lymphocytes. In a study involving systemic knockout combined with competitive transplantation, it was found that Prdm1 promotes NK cell maturation and the expression of Gzmb. On the contrary, the same study also found that NK cells with Prdm1 deficiency exhibit heightened proliferation, increased survival, enhanced migratory abilities toward tumors, and greater cytotoxicity against subcutaneously implanted RMAS tumors (*Kallies et al., 2011*). Using Ncr1-driven conditional knockout transgenic mice, which specifically delete *Prdm1* in group 1 ILCs, we not only validated *Prdm1*'s positive regulation of NK cell maturation, but also demonstrated its indispensable role in NK cell anti-tumor activity. The compromised mitochondrial function and reduced IFN-γ, granzyme B, and perforin production appear to be potential contributing factors to Prdm1-mediated NK cell cancer surveillance. Reduction of CX3CR1+ NK cells in multiple tissues, and decreased expression of *Cx3cr1* and *Cxcr6* was observed in *Prdm1* cko splenic cNK cells (*Figure 3F*), both of which are essential for NK cells egressing from bone marrow (*Sciumè et al., 2011*; *Chea et al., 2015*). Our results not only confirmed a decrease in cytotoxic molecules in *Prdm1*-deficient NK cells (*Kallies et al., 2011*) but also showed that the reduction in Gzmb and perforin is not solely attributable to the diminished maturation of these cells. Mature NK cells in bone marrow obtained the expression of CX3CR1 and acquired ability to enter the circulation (*Sciumè et al., 2011*; *Chea et al., 2015*). The quantity of cNK cells increased exclusively in the bone marrow, with reductions observed in all other tissues (*Figure 1G*), indicating *Prdm1* might regulate chemokine receptors to facilitate the egression of cNK cells from the bone marrow to peripheral tissues. In addition, higher expression of *Cxcr6* compared to cNK cells is also a key factor for liver tissue residence of ILC1s (*Hudspeth et al., 2016*). Decreased expression of *Cxcr6* in *Prdm1* deficient group 1 ILCs may also contribute to the balance shift towards cNK cells. Furthermore, although both liver NK cells and liver ILC1s require Prdm1 to maintain their quantity, liver ILC1s demonstrate a more pronounced dependency on Prdm1. However, it is currently widely believed that liver NK cells and liver ILC1s originate from different progenitors. It is worth noting that while we observed changes in the NK and ILC1 proportions after *Prdm1* knockout, our data does not support the hypothesis that Prdm1 affects progenitor differentiation decisions, thereby influencing the fate selection of NK and ILC1. Further research may be needed to elucidate how Prdm1 regulates the balance between NK cells and ILC1s.

The scRNA-seq analysis reveals that both liver cNK cells and ILC1s can be further divided into three subgroups based on their gene expression patterns. JunB is a crucial transcriptional factor for the cytotoxic function of CD8+ T cells and NK cells. However, excessive *Junb* expression has been found to promote T cell exhaustion (*Lynn et al., 2019*). Our previous study showed that as NK cells mature, the expression level of *Prdm1* increased while the expression level of *Junb* gradually decreased (*Wang et al., 2018*). The current study demonstrated that in NK cells, the expression level of *Junb* significantly increases upon the deletion of Prdm1, indicating that *Junb* expression is suppressed by Prdm1. As *Junb* expression decreases with NK cell maturation, and it is inhibited by the gradually increasing Prdm1 during maturation. This implies that constraining *Junb* expression is likely a fundamental prerequisite for NK cell maturation. However, the precise mechanism by which Prdm1 downregulates *Junb* in NK cells still needs further research. Furthermore, *Junb* high NK cells exhibit lower expression levels of cytotoxic genes and reduced mitochondrial-related signaling pathways. Mitochondria are pivotal organelles crucial for cellular metabolism. Disruptions in mitochondrial function have been linked to T Cell exhaustion, attributed to glycolytic reprogramming (*Wu et al., 2023*). Similarly, mitochondrial fragmentation has been closely associated with NK cell exhaustion (*Zheng et al., 2019*). However, the concept of NK cell exhaustion isn't as firmly established as it is for T cells. Exhausted NK cells should primarily exhibit diminished functions. This is characterized by a diminished ability to destroy tumor cells, a reduced capability to activate other components of the immune system, and compromised proliferation and survival rates. Additionally, this reduced functionality is associated with a decline in the expression of molecules responsible for cytotoxic activity, lower production of IFN-γ, and metabolic disturbances that may arise from mitochondrial dysfunction. While our current data is not sufficient to definitively classify these cells as exhausted NK cells, it supports

that a subpopulation, referred to *Junb* high cluster, demonstrates an exhaustion-like phenotype. The significant increase in this cell population following *Prdm1* knockout in NK cells may potentially be one of the reasons why *Prdm1* cko mice lose their tumor-killing capacity. Whether the excessive expression of JunB in NK cells is also a contributing factor to their exhaustion, similar to T cells (*Lynn et al., 2019*), requires further investigation.

The scRNA-seq data revealed that *Prdm1* plays distinct roles in regulating cNK cells and ILC1s despite being required for the quantity of both lineages. Specifically, *Prdm1* appears to be more involved in promoting the resistance against exhaustion in cNK cells, whereas in ILC1s, it may play a role in the plasticity between ILC1s and ILC3s. In both our previous study and a study by Colonna et al. (*Wang et al., 2018*; *Cheng et al., 2021*), it was demonstrated that *Smad4*, a transcriptional factor involved in TGF-β signal pathway, upregulated *Prdm1* in NK cells and depletion of *Smad4* resulted in a decreased ratio of NK cells to ILC1s in the liver. However, knocking out *Prdm1* in Ncr1+ cells increased the ratio of NK cells to ILC1s in liver group 1 ILCs (*Figure 2A*). These findings suggest the possibility of a Smad4-independent pathway through which *Prdm1* promotes the maintenance of ILC1s, or that *Prdm1* plays a more significant role in maintaining ILC1s compared to its role in NK cells.

Previous studies have identified Hobit and Prdm1 as central regulators instructing tissue-dependent programs and retention of diverse tissue-resident lymphocytes (*Mackay et al., 2016*; *Friedrich et al., 2021*; *Yomogida et al., 2021*). Liver ILC1s required Hobit, but not necessary for cNK cells (*Ducimetière et al., 2021*). Expression of Prdm1 was remarkably higher in cNK cells versus ILC1s (*Mackay et al., 2016*). While in our study, cNK cells and liver ILC1s reduced simultaneously in *Prdm1* cko mice, and even more significant in ILC1s. This indicates that while Prdm1 is expressed at lower levels in ILC1s, its role in preserving the quantity of ILC1s may be more crucial. Thus, Prdm1 and Hobit may have parallel program in instructing ILC1s functional development and maturation. Prdm1 and Hobit directly bound and repressed *Tcf7* (*Mackay et al., 2016*), which encoded TCF-1, a TF binding and limiting the activity of Gzmb regulatory element (*Jeevan-Raj et al., 2017*). Gzmb has been demonstrated directly bound and activated by Junb in NK cells, which suggested Gzmb expression regulated by multiple Prdm1/Hobit downstream signals (*Wang et al., 2018*). In human T cells, binding motif of JUNB was enriched in the binding sites of PRDM1 (*Guo et al., 2022*), indicating the essential role of PRDM1-JUNB axis during NK cell and T cell development. In *Prdm1* deficient NK cells, we noted a decrease in Gzmb levels alongside with an elevation in Junb expression. This indicates that Prdm1 not only facilitates the expression of Gzmb in NK cells but also suppresses Junb expression. Given that Junb is recognized as a positive regulator of Gzmb (*Babichuk and Bleackley, 1997*), this presents a complex interplay that seems contradictory. Therefore, it is imperative to develop a theoretical framework to comprehensively understand and interpret this paradoxical relationship. Here, we hypothesize that during the early stages of NK cell development, JunB may enhance the expression of certain molecules associated with cytotoxicity, thereby aiding NK cells in acquiring the capacity to eliminate target cells. At these initial stages, Prdm1 level are comparatively low, thus exerting a weaker inhibitory effect on JunB. As NK cells mature, there is a progressive increase in Prdm1 level, which then exerts a stronger inhibitory influence on JunB, contributing to the reduced JunB level in fully matured NK cells. Consequently, this reduces the potential for NK cell exhaustion caused by elevated levels of JunB.

Chronic inflammation is a crucial factor in promoting tumorigenesis, and macrophages play a significant role in this process. Macrophages interact with both cNK cells and ILC1s. However, the TFs that regulate these interactions are poorly understood. Fortunately, recent advances in scRNA-seq technology and the CellChat software tool *Jin et al., 2021* have allowed us to gain a better understanding of the *Prdm1* signaling pathway in group 1 ILCs and its impact on macrophages at the transcriptional level. Increased metastasis in *Prdm1* cko mice may be due to a decrease in the killing ability of NK cells, making them more prone to exhaustion, or it could be due to abnormalities in group 1 ILCs, leading to a decrease in the anti-tumor abilities of macrophages and an enhancement of their pro-tumor capabilities. It is worth noting that normal ILC1-macrophage interactions are more prevalent than the interaction between NK cells and macrophages (*Figure 6—figure supplement 2*). We also found that CXCL signaling-based interaction remarkably diminished in *Prdm1* cko mice, which suggested CXCL-CXCR may contribute to keep the sufficient interaction between group 1 ILC and macrophage. Specifically, *Prdm1* in group 1 ILCs may be critical in preventing the overactivation of macrophages that can lead to cancer development. Increased population of both MDMs and KCs in *Prdm1* cko mice, as well

as different distribution of macrophage clusters, indicating the homeostasis of macrophages require environment with functional cNK cells and ILC1s. Although not all macrophage phenotypes have been verified in this study, the present research serves primarily to offer initial insights and preliminary data for investigating the interactions between group 1 ILCs and macrophages. It aims to inspire further research into the role of transcription factors within the liver and the cancer microenvironment.

While our findings underscore the importance of Prdm1 in liver cNK cells and ILC1s tumor immune surveillance, it does not be validated in human NK cells, whereas previous studies have found that PRDM1 might inhibit the proliferation and function of human NK cells, or human NK cell derived cancer cells (*Smith et al., 2010*; *Küçük et al., 2011*). Furthermore, we did not provide an in-depth evaluation in multiple tumor models. Further research may provide deeper insight into the role of PRDM1 in the anti-tumor function of human NK cells, enabling a more direct investigation of its application in cancer therapies. Given its important role in preserving liver cNK cells and ILC1s functional heterogeneity, enhancing Prdm1 function in human NK cells could potentially be a strategy to promote NK-cell-based immunotherapy for cancer.

## Methods

### Mice

*Prdm1*$^{fl/fl}$ mice were purchased from The Jackson Laboratory. *Ncr1-iCre* and *B2m*$^{-/-}$ mice were purchased from Shanghai Model Organisms Center, Inc. Cre is targeted to the *Ncr1* locus. Six- to twelve-week-old littermates were used for the experiment. All animal experiments were approved by Tianjin University Institutional Animal Care and Use Committee.

### Cell lines

B16F10 melanoma cells were obtained from ATCC (CRL-6475) and negative for mycoplasma contamination.

### Experimental metastasis model

For lung metastasis model, $0.3×10^6$ B16F10 cells were intravenous injected into mice. Three weeks later, mice were euthanized for analysis. For liver metastasis model, mice were inoculation with $0.5×10^6$ B16F10 via intrasplenic injection. Three weeks later, mice were euthanized for analysis. Lung and liver from tumor-bearing mice were fixed in 10% formalin and embedded in paraffin. Sections were stained with H&E.

### In vivo cytotoxicity assay

Donor splenocytes harvested from B2m deficient (*B2m*$^{-/-}$) mice were labeled with 5 μM CFDA-SE. Donor splenocytes harvested from B2m-adequate (*B2m*$^{+/+}$) mice were labeled with 5 μM eF670. Labeled splenocytes from two mouse strains were mixed in a 1:1 ratio, and $1×10^7$ cells in total were injected i.v. into *Prdm1*$^{+/+}$ and *Prdm1* cko mice. One day after administration, spleen and liver cells were isolated from recipient mice, and the population of labeled cells was analyzed by flow cytometry. Rejection % was quantified according to the following formula:

$1-(\%B2m^{+/+}/\%B2m^{-/-})_{Prdm1}{}^{+/+}{}_{or\ Prdm1}$ cko ×100%.

### Cell isolation

Mice were perfused with PBS by portal vein puncture before harvesting tissues. Liver and lung was digested with 0.05% collagenase II for 30 min and filtered through 70 μm cell strainers, and mononuclear cells were isolated after subjected to density gradient using 30% and 70% percoll. Spleen were also removed and pressed through 70 μm filters to obtain splenocytes. Peripheral blood mononuclear cells were obtained from peripheral blood after lysis of red blood cells (Biolegend, 420301). Flushing femurs and mechanical disruption of inguinal lymph nodes were performed to obtain cells from bone marrow and lymph nodes.

### Real-time RT-PCR

RNA was extracted from FACS-sorted NK cells or splenocytes using RNASimple Total RNA Kit (TIANGEN Biotech, 4992858) and subsequently reverse transcribed to cDNA with SuperScript VILO

Master Mix (Thermo Fisher Scientific, 11755050) according to manufacturer's instructions. qPCR was performed with SYBR Green Mix (Thermo Fisher Scientific, A25742) and CFX Opus 96 Real-Time PCR System (Bio-Rad). The relative mRNA expression level was calculated using $2^{-ddCt}$ method. Primer sequences: *Prdm1*: 5'-CAGAAACACTACTTGGTACA-3'; 5'-GATTGCTTGTGCTGCTAA-3'.

## Flow cytometry

Flow cytometry and cell sorting were performed with a Cytoflex S/SRT (Beckman Coulter). The following antibodies were used (all purchased from BioLegend unless otherwise indicated): CD45-PE-Cy7 (catalog 103114, clone 30-F11); CD3ε-PerCP-Cy5.5 (catalog 100327, clone 145–2 C11); NK1.1-BV421 (catalog 108741, clone PK136); CD335-AF647 (catalog 560755, clone 29A1.4, BD Bioscience); CD49a-PE (catalog 562115, clone Ha31/8, BD Bioscience); CD49b-FITC (catalog 108906, clone DX5); CD27-BV510 (catalog 124229, clone LG.3A10); CD11b-AF700 (catalog 101222, clone M1/70); KLRG1-APC (catalog 561620, clone 2F1, BD Bioscience); CD49a-BV421 (catalog 740046, clone Ha31/8, BD Bioscience); IFN-γ-PE (catalog 505807, clone XMG1.2); Granzyme B (catalog 372207, clone QA16A02); Perforin (catalog 154305, clone S16009A); CD49b-APC-Cy7 (catalog 108919, clone DX5); IgG-PE (catalog 402203, clone 27–35); CD335-BV510 (catalog 137623, clone 29A1.4);; CD3ε-APC-Cy7 (catalog 100330, clone 145–2 C11); CD11b-BV421 (catalog 101235, clone M1/70); CD3ε-BV421 (catalog 100335, clone 145–2 C11); CD3ε-FITC (catalog 553061, clone 145–2 C11, BD Bioscience); CX3CR1-PE (catalog 149005, clone SA011F11); Ly6G-PerCP-Cy5.5 (catalog 127615, clone 1A8); Ly6C-BV510 (catalog 108437, clone RB6-8C5); F4/80-APC-Cy7 (catalog 123118, clone BM8). For mitochondrial metabolic assay, fresh cells were incubated in 37°C media for 30 min with 100 nM MitoTracker Green (catalog M7514, Invitrogen), 100 nM TMRM (catalog T668; Invitrogen), and 10 uM MitoSOX Red (catalog M36008; Invitrogen), respectively. Surface-stained after washing with PBS and then detected by flow cytometry. For intracellular IFN-γ staining, Cells freshly obtained from liver and spleen were stimulated 12 hr with or without cytokine. GolgiStop (BD Biosciences) was added 4 hr before intracellular staining.

## Bulk RNA sequencing

Total RNA from FACS sorted splenic NK cells of *Prdm1*⁺/⁺ and *Prdm1* cko mice was extracted by TRIzol reagent (Invitrogen), then reverse transcribed into cDNA. Library construction was prepared using Illumina mRNA Library kit, and sequencing was performed by the BGISEQ-500. Standard methods were used to analyze the RNA-seq data, including aligning the reads to the genome by HISAT2 (v2.1.0) (*Kim et al., 2015*), and gene expression values (Counts) were calculated using RSEM (v1.3.1) (*Li and Dewey, 2011*). DEGs were identified using DEseq2 (v1.4.5) (*Love et al., 2014*) with a cutoff of $\log_2$(-fold change)>0.5 and p<0.05. The 'clusterProfiler' package (v4.4.4) (*Wu et al., 2021*) and gene sets from molecular signatures database (MSigDB) were used for GSEA and GO analysis. The heatmap was plotted using the 'Pheatmap' package (v1.0.12).

## Single-cell RNA sequencing

FACS-sorted liver CD45⁺ cells with more than 80% cell viability were used for library preparation. Each sample contained cells from three *Prdm1*⁺/⁺ or *Prdm1* cko mice. Gel Bead-in-Emulsions (GEMs) were generated using the 10 X Genomics Chromium system, which combinates Master Mix, Single Cell 3' v3.1 Gel Beads, and Partitioning Oil with single cells. GEMs were mixed with cell lysate and reverse transcription reagent to produce full-length cDNA. After incubation, the GEMs were broken, and recovered cDNA were amplified via PCR. Fragmentation, End repair, A-tailing, and Adaptor Ligation were performed to obtain final libraries, which contain P5 and P7 sequences. The 3' library was sequenced on Novaseq 6000 with approximately 50 k read pairs/cell sequencing depth. The 'Seurat' R package (v4.2.0) (*Hao et al., 2021*) was used for data analysis. Initial quality control was performed to filter out the low-quality cells and cell doublets. Cells with 200–5500 expressed genes and no more than 10% mitochondrial genes were considered for high-quality. Doublets were filtered with the 'scDblFinder' Package (v1.10.0) (*Germain et al., 2021*). After quality control, we totally recovered 6161 cells and 4817 cells from *Prdm1*⁺/⁺ and *Prdm1* cko mice, respectively. Principal component analysis (PCA) was used for cluster analysis. The first 15 PCs were picked for clustering and further visualized by UMAP. Clusters-specific marker was defined using the 'FindAllMarkers' function, and clusters were manually annotated based on the top 30 or 15 markers. The 'clusterProfiler' package and gene

set from MSigDB were used for GSEA and GO analysis. 'CellChat' package (v1.4.0) (*Jin et al., 2021*) was utilized to predict the cell-to-cell communication from scRNA-seq data.

## TCGA datasets assay

The normalized gene expression and survival datasets of cancer patients collected in The Cancer Genome Atlas (TCGA) were downloaded from UCSC Xena (http://xena.ucsc.edu/) (*Caicedo et al., 2020*). NK cell-associated genes including *CD160*, *CD244*, *CTSW*, *FASLG*, *GZMA*, *GZMB*, *GZMH*, *IL18RAP*, *IL2RB*, *KIR2DL4*, *KLRB1*, *KLRC3*, *KLRD1*, *KLRF1*, *KLRK1*, *NCR1*, *NKG7*, *PRF1*, *XCL1*, *XCL2*, according to the previous study (*Cursons et al., 2019*). NK-cell-associated genes, together with *PRDM1*, constitute the *NK-PRDM1* signature in this study. The mean expression of per genes was ordered from high-to-low and plotted by heatmap using the 'Pheatmap' package. The overall survival of patients in the high and low expression of *NK-PRDM1* signature was selected for analysis. Kaplan-Meier curves were plotted by GraphPad Prism.

## Statistics

For experiment results, two-tailed t tests were used to measure the continuous and normally distributed between the two independent groups. Paired t-tests were used to determine the statistical significance between two paired groups. Log-rank tests were used to compare the overall survival distribution between the two groups of patients. A p value less than 0.05 was considered significant and data were presented as mean ± SEM.

## Study approval

All animal experiments were approved by The Tianjin University Animal Care and Use Committee. No human subjects were performed in this study (protocol: 00000000202010100023).

## Acknowledgements

This work was supported by National Key Research and Development Plan of China (2022YFF1202901); National Natural Science Foundation of China (82372801); The Zhejiang Provincial Natural Science Foundation of China (LY21H150002); Natural Science Foundation of Tianjin (23JCYBJC01560, 23JCYBJC01370, 21JCZDJC00430); and Science and Technology Planning Project of Tianjin Municipal Education Commission (2022YGYB14).

## Additional information

### Funding

| Funder | Grant reference number | Author |
|---|---|---|
| National Key Research and Development Plan of China | 2022YFF1202901 | Youwei Wang |
| National Natural Science Foundation of China | 82372801 | Youwei Wang |
| The Zhejiang Provincial Natural Science Foundation of China | LY21H150002 | Zhouxin Yang |
| Natural Science Foundation of Tianjin | 23JCYBJC01560 | Xue Li Huaiyong Chen Youwei Wang |
| Science and Technology Planning Project of Tianjin Municipal Education Commission | 2022YGYB14 | Xue Li |

| Funder | Grant reference number | Author |
|---|---|---|
| Natural Science Foundation of Tianjin | 23JCYBJC01370 | Xue Li<br>Huaiyong Chen<br>Youwei Wang |
| Natural Science Foundation of Tianjin | 21JCZDJC00430 | Xue Li<br>Huaiyong Chen<br>Youwei Wang |

The funders had no role in study design, data collection and interpretation, or the decision to submit the work for publication.

### Author contributions

Jitian He, Conceptualization, Resources, Data curation, Formal analysis, Supervision, Validation, Investigation, Visualization, Methodology, Writing – original draft, Writing – review and editing; Le Gao, Data curation, Formal analysis, Validation, Investigation, Methodology, Writing – original draft; Peiying Wang, Conceptualization, Data curation, Formal analysis, Validation, Investigation, Methodology; Wing Keung Chan, Xiao-Hong Li, Project administration; Yiran Zheng, Investigation, Methodology; Yumo Zhang, Resources, Software, Validation, Visualization; Jiaman Sun, Visualization, Methodology; Xue Li, Data curation, Formal analysis, Funding acquisition; Jiming Wang, Conceptualization; Huaiyong Chen, Supervision, Project administration; Zhouxin Yang, Supervision, Funding acquisition, Project administration; Youwei Wang, Conceptualization, Resources, Data curation, Software, Formal analysis, Supervision, Funding acquisition, Validation, Investigation, Visualization, Methodology, Writing – original draft, Project administration, Writing – review and editing

### Author ORCIDs

Jitian He ⓘ http://orcid.org/0009-0008-8425-3481
Wing Keung Chan ⓘ https://orcid.org/0000-0002-5257-1521
Youwei Wang ⓘ https://orcid.org/0000-0002-1736-1203

### Ethics

All animal experiments were approved by The Tianjin University Animal Care and Use Committee (protocol: 00000000202010100023).

Reviewer #1 (Public Review): https://doi.org/10.7554/eLife.92948.3.sa1
Reviewer #2 (Public Review): https://doi.org/10.7554/eLife.92948.3.sa2
Author response https://doi.org/10.7554/eLife.92948.3.sa3

# Additional files

### Supplementary files
• MDAR checklist

### Data availability

Sequencing data have been deposited in Gene Expression Omnibus (GEO) dataset under accession codes GSE271233 (scRNA-seq) and GSE271380 (RNA-seq). All other data are available within the article and its supplementary information.

The following datasets were generated:

| Author(s) | Year | Dataset title | Dataset URL | Database and Identifier |
|---|---|---|---|---|
| He J, Gao L, Wang P, Wang Y | 2024 | Prdm1 Positively Regulates Liver Group 1 ILCs Cancer Immune Surveillance and Preserves Functional Heterogeneity | https://www.ncbi. nlm.nih.gov/geo/ query/acc.cgi?acc= GSE271233 | NCBI Gene Expression Omnibus, GSE271233 |
| He J, Gao L, Wang P, Wang Y | 2024 | Prdm1 Positively Regulates Liver Group 1 ILCs Cancer Immune Surveillance and Preserves Functional Heterogeneity | https://www.ncbi. nlm.nih.gov/geo/ query/acc.cgi?acc= GSE271380 | NCBI Gene Expression Omnibus, GSE271380 |

The following previously published datasets were used:

| Author(s) | Year | Dataset title | Dataset URL | Database and Identifier |
|---|---|---|---|---|
| Yomogida K, Colonna M | 2021 | Hobit confers tissue dependent programs to type 1 innate lymphoid cells | https://www.ncbi. nlm.nih.gov/geo/ query/acc.cgi?acc= GSE185346 | NCBI Gene Expression Omnibus, GSE185346 |
| Friedrich C, Taggenbrock R, Doucet-Ladevèze R, Golda G, Moenius R, Arampatzi P, Kragten NA, Kreymborg N, Kastenmueller W, Saliba AE, Grün D, van Gisbergen KP, Gasteiger G | 2021 | Effector differentiation downstream of lineage commitment in ILC1 is driven by Hobit across tissues | https://www.ncbi. nlm.nih.gov/geo/ query/acc.cgi?acc= GSE163452 | NCBI Gene Expression Omnibus, GSE163452 |
| Flommersfeld S, Böttcher JP, Ersching J, Flossdorf M, Meiser P, Pachmayr LO, Leube J, Hensel I, Jarosch S, Zhang Q, Chaudhry MZ, Andrä I, Schiemann M, Busch DH, Cicin-Sain L, Sun JC, Gasteiger G, Victora GD, Höfer T, Buchholz VR, Grassmann S | 2021 | Fate mapping of single NK cells identifies a type 1 innate lymphoid-like lineage that bridges innate and adaptive recognition of viral infection [bulk_RNAseq_steady_state] | https://www.ncbi. nlm.nih.gov/geo/ query/acc.cgi?acc= GSE180973 | NCBI Gene Expression Omnibus, GSE180973 |

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
