## [Editor Report · eLife assessment]

The authors investigated the requirement and function of Blimp1/Prdm1 in murine natural killer (NK) cells and the ILC1 lineage of innate lymphoid cells, using a conditional knockout model. The single-cell mRNA-seq data provided here represent a **valuable** resource for the community, but the lack of mechanistic investigations leaves the study partially **incomplete**. The work will be of interest to the fields of innate lymphoid cell biology and tissue immunology.

---

## [Referee Report · Reviewer #1 (Public Review)]

He et al. investigate the requirement and function of Blimp1 (encoded by Prdm1) in murine NK cells and ILC1. Employing a conditional knockout mouse model (Prdm1flox x Ncr1cre), the authors describe impaired abundance and maturation of Prdm1-deficient NK cells and ILC1 in different tissues. Blimp1-deficient NK cells have reduced expression of cytotoxic molecules (Gzmb, Prf1) and, in some instances, Ifng production, and Prdm1flox x Ncr1cre mice show impaired tumor control in experimental metastasis models. Using single cell RNA sequencing analysis, the authors propose that Prdm1 regulates JunB expression and NK cell maturation. Based on in silico analyses, the authors suggest manifold intercellular communication between NK/ILC1 and macrophages. Without following up on any of these potentially interesting suggestions, the authors conclude their study reiterating that Prdm1 regulates IFNg-production of tumor-infiltrating NK cells and ILC1.

Many of the reported functions of Blimp1 in NK cells have previously been identified using a mixed-chimera strategy comparing Prdm1 WT and KO NK cells (Kallies et al., Blood 2011). Here, the authors expand on these findings using a conditional model to delete Prdm1 in NK/ILC1 and single cell sequencing, and provide a more refined analysis of the functions of Blimp1 in these cells. Cell-chat analysis suggests close interactions fo Blimp-dependent NK/ILC1 subsets with hepatic macrophages, but these suggestions are not followed up by experiments. Potentially interesting differences in the macrophage compartment of Ncr1-Cre x Prdm1-fl/fl mice are suggested by the scc-RNA-Seq data, but are not validated e.g. by FACS. The study falls short in providing new mechanistic insights. Nevertheless, it is an interesting confirmation of "old" suggestions in a more refined setting, and the provided single-cell mRNA-Seq data represents a potentially valuable resource for the community.

---

## [Referee Report · Reviewer #2 (Public Review)]

He and colleagues aimed to elucidate the role of the transcription factor Prdm1 in liver Type 1 ILCs (innate lymphoid cells), focusing on its regulatory mechanisms and potential implications for developing innovative immune therapy strategies against liver cancer.

Strengths:

The study effectively integrates omics analyses and cytometry to explore Prdm1's impact on the cellular composition and immune regulation within the liver, providing a comprehensive view of its biological role. Employing a conditional knockout mouse model adds specificity to their experiments, allowing for precise manipulation of the Prdm1 gene.

Weaknesses:

The study predominantly relies on limited mouse models, which may not fully represent the complexity of Type 1 ILC behavior across different cancer types. Some experimental designs, such as the limited in vitro killing assessments, and additional human data could be expanded to strengthen the findings and their interpretation.

The authors have demonstrated that Prdm1 plays a critical role in the function of NK cells and ILC1s within the liver, particularly in the context of tumor resistance. However, due to the use of specific disease models and lack of direct human data, the application of these findings to clinical settings remains speculative. While the study advances our understanding of liver ILC biology, further research is necessary to validate these effects in human systems and across more diverse cancer models.

Discussion on impact and utility:

This study contributes significantly to the field of immunology and cancer therapy by revealing potential new targets for immunotherapy of liver cancer. The methods and data provided could serve as a valuable resource for further research aimed at enhancing immune-based cancer treatments.

Additional Context for Interpretation:

Understanding the role of Prdm1 in the broader context of immune cell regulation and its interaction with other cellular components in the tumor microenvironment could be crucial. Further studies should explore the dynamic between Prdm1 expression, NK cell functionality, and tumor resistance mechanisms to fully harness the therapeutic potential of targeting this pathway in liver cancer.

---

## [Author Response]

The following is the authors’ response to the previous reviews.

**Reviewer 1 (Public Review):**
He et al. investigate the requirement and function of Blimp1 (encoded by Prdm1) in murine NK cells and ILC1. Employing a conditional knockout mouse model (Prdm1flox x Ncr1cre), the authors describe impaired abundance and maturation of Prdm1-deficient NK cells and ILC1 in different tissues. Blimp1-deficient NK cells have reduced expression of cytotoxic molecules (Gzmb, Prf1) and, in some instances, Ifng production, and Prdm1flox x Ncr1cre mice show impaired tumor control in experimental metastasis models. Using single-cell RNA sequencing analysis, the authors propose that Prdm1 regulates JunB expression and NK cell maturation. Based on in silico analyses, the authors suggest manifold intercellular communication between NK/ILC1 and macrophages. Without following up on any of these potentially interesting suggestions, the authors conclude their study reiterating that Prdm1 regulates IFNg-production of tumor-infiltrating NK cells and ILC1. Many of the reported functions of Blimp1 in NK cells have previously been identified using a mixed-chimera strategy comparing Prdm1 WT and KO NK cells (Kallies et al., Blood 2011). Here, the authors expand on these findings using a conditional model to delete Prdm1 in NK/ILC1 and single-cell sequencing and provide a more refined analysis of the functions of Blimp1 in these cells. Cell-chat analysis suggests close interactions of Blimp-dependent NK/ILC1 subsets with hepatic macrophages, but these suggestions are not followed up by experiments. Potentially interesting differences in the macrophage compartment of Ncr1-Cre x Prdm1-fl/fl mice are suggested by the scRNA-Seq data but are not validated e.g. by FACS. The study falls short in providing new mechanistic insights. Nevertheless, it is an interesting confirmation of "old" suggestions in a more refined setting, and the provided single-cell mRNA-Seq data represents a potentially valuable resource for the community. There are some control analyses that are required to support the conclusions of the authors, and I have a few suggestions that would help to improve the manuscript.

We sincerely appreciate your careful review and insightful feedback on our manuscript. We have carefully considered your comments and present the results of new experiments conducted in response to your suggestions. Please find the detailed responses below.

Major commentsComment 1: The authors do not control for the potential effects of Cre expression. Expression of Cre from within the Ncr1 locus (using the mouse model established by Narni-Mancinelli et al.) has significant effects on NK cells and especially ILC1s reducing their frequency and absolute numbers and altering their functionality. The authors should characterize the Ncr1cre mice used here (developed by Shanghai Model Organism Center) in this regard and should use proper controls (Ncr1Cre+ Prdm1wt/wt as control for Ncr1Cre+ Prdm1fl/fl, instead of WT littermates) for all of their key data, e.g. those depicted in Fig 1FG, 2ADFH, 7D, S2,3,4.

Response 1: This is a very insightful question that has posed a challenge for many researchers, including us, engaged in conditional knockout studies. The expression of Cre and the insertion of loxP sequences both have the potential to influence gene expression. This is because the region where loxP is inserted may contain regulatory sequences for the gene of interest. Ncr1-Cre is a frequently used transgenic mouse model in our laboratory. In our prior research, we also had concerns about the possible impact of Cre on NKp46 expression, which could lead to a decline in NK cell function. Therefore, in our previous studies focused on Smad4 expression in NK cells, we conducted similar experiments. In Figure 6 of our published paper in the Journal of Clinical Investigation (Wang et al., *J Clin Invest*, 2018), we compared NKp46-iCreTgfbr2fl/flSmad4fl/WT with NKp46-iCreTgfbr2fl/flSmad4fl/fl. Although the primary purpose is to establish Smad4's independence from TGF-β, it also allows for a comparison between Smad4fl/fl and Smad4fl/WT in the presence of Cre. In the critical phenotype we assessed, NKp46-iCreTgfbr2fl/flSmad4fl/fl (compared with NKp46-iCreTgfbr2fl/flSmad4fl/WT) exhibited the same phenotype as NKp46-iCreSmad4fl/fl (compared with NKp46WTSmad4fl/fl). This suggests that Cre's influence on NK cells may be within a reasonable and controllable range. Furthermore, in contrast to the decrease in Ncr1 expression caused by Cre, the reduction in the expression levels of genes targeted by Loxp knockout, such as Prdm1 in this study (Figure 1 E), is more significant. Therefore, with the current techniques and research methods, we believe that the data provided in this study can support the role of Prdm1 in

NK cells.

Comment 2: Several of the phenotypic findings on NK cells have been described before by Kallies et al. in 2011 (Ref 29), although using a different genetic Prdm1-ablation model (Prdm1-GFP/GFP knockin/knockout model). This study reported impaired NK cell maturation, reduced Gzmb expression, impaired in vivo cytotoxicity against subcutaneous RMA-S cells, impaired in vitro proliferation, comparable in vitro killing, increase in BM NK cell numbers. The authors should discuss/mention this more prominently in their manuscript, and highlight where they confirm or refine these previous findings, and where they actually provide new information.

Response 2: We appreciate your valuable suggestions. The article you referred to, published in *Blood*, is indeed an excellent work. While we had cited this article, our discussion regarding its specific content was limited. Based on your advice, we have made revisions and included the following content in our discussion section (page 24; line 489-493):

“In a study involving systemic knockout combined with competitive transplantation, it was found that Prdm1 promotes NK cell maturation and the expression of Gzmb. On the contrary, the same study also found that NK cells with Prdm1 deficiency exhibit heightened proliferation, increased survival, enhanced migratory abilities towards tumors, and greater cytotoxicity against subcutaneously implanted RMAS tumors (31).”.

Comment 3: What is the reason to refer to the enriched cluster in Blimp1-deficient NK cells as "Junbhi"? There is no follow-up for a function of Junb, and there are many other genes upregulated in these cells. Most critically, these cells seem to represent exactly the c-Kithi cells that Kallies et al. already showed and discussed in their paper. The authors should stain for Kit, and also refer to this. Also, MacKay et al. performed Blimp1-Chip-Seq (in T cells), maybe it would be interesting to check to which of the identified DEGs Blimp1 can bind.

Response 3: We appreciate the suggestion from the reviewer. We think a gene that supports the development of lymphocytes doesn't necessarily positively regulate their function. For example, JunB is essential for T cell development but can also induce T cell exhaustion (Lynn et al., *Nature*. 2019). Therefore, while Prdm1 has been shown to promote NK cell development, it cannot be assumed that it always positively regulates NK cell function, especially for anti-cancer immune surveillance. In this respect, we try to find a driving-factor of the impaired anti-tumor ability of *Prdm1*Δ*Ncr1* NK cells. Although there are many other genes upregulated in this cluster (e.g. Kit), JunB attracts more our interest of its potential for regulating NK cells functions in cancer, whereas c-Kit is more likely a marker of NK cells maturation, which has been well-demonstrated by Kallies et al. and other studies. Our previous studies also showed that the expression of c-kit was decreased in mature NK cells, compared immature NK cells (Wang et al., *J Clin Invest*, 2018).

The lack of following experiments of Junb is because we cannot find valuable surface markers to investigate the follow-up function of _Junb_hi cNK cluster. If we use intracellular markers, it is more likely an analysis of gene expression pattern, which has been well-described in our RNA-seq data. As we describe above, our study did not aim to further investigate the role of prdm1 in NK cells maturation, as the c-Kit expression was upregulated in Prdm1-kncok NK cells and correlated with NK cell maturation, which has been validated by Kallies et al..

We also have discussed the potential DEGs that could be bound and regulated by Prdm1 in our revised manuscript (page 27-28; line 561-571):

“Prdm1 and Hobit directly bound and repressed *Tcf7* (18), which encoded TCF-1, a TF binding and limiting the activity of Gzmb regulatory element (69). Gzmb has been demonstrated directly bound and activated by Junb in NK cells, which suggested Gzmb expression regulated by multiple Prdm1/Hobit downstream signals (26). In human T cells, binding motif of JUNB was enriched in the binding sites of PRDM1 (70), indicating the essential role of PRDM1-JUNB axis during NK cell and T cell development. In NK cells deficient in Prdm1 expression, we noted a decrease in Gzmb levels alongside with an elevation in Junb expression. This indicates that Prdm1 not only facilitates the expression of Gzmb in NK cells but also suppresses Junb expression. Given that Junb is recognized as a positive regulator of Gzmb (71), this presents a complex interplay that seems contradictory. Therefore, it is imperative to develop a theoretical framework to comprehensively understand and interpret this paradoxical relationship.”.

Comment 4: cNK cells are considered circulating cells, that transiently pass through the liver.

Previous studies have suggested almost identical gene expression patterns in hepatic and splenic NK cells. In functional tests, they often "perform" identically. I am therefore a bit surprised that the authors find a differential dependency of Blimp1 for the IFNg production of splenic (no role of Blimp1) versus hepatic (Blimp1 regulating IFNg production) NK cells (Fig S3). Do the authors have any suggestions on that? The analyses are performed by 12+4h stimulations with IL12/18, which could involve the effects of altered bystander cells (as suggested by Figure 6). Therefore, these analyses should be provided upon standard 4h stimulations with IL12/18 and also with PMA/I under BFA. Note: liver and splenic cNK cells look quite different in the chosen histograms in Figures 7 A, B, C, yet there is massive variability in these analyses - is there any systematic/technical problem?

Response 4: We appreciate the valuable suggestion from the reviewer. Studies have suggested that, at the gene expression or transcriptomic level, liver NK cells exhibit more similarity to splenic NK cells while displaying greater divergence from liver ILC1s. However, we do not think that splenic NK cells or peripheral blood NK cells (which are more abundant in circulation) are entirely indistinguishable from liver NK cells. Notably, there are substantial differences in their maturity levels, with liver NK cells being more mature. Since we are examining the protein levels, a 4-hour stimulation period may not fully capture these distinctions. Even when considering the potential impact of bystander cells, the experimental design specifically targets Prdm1 knockout within NK cells, ensuring that the study accurately elucidates the role of Prdm1 in NK cells. For each experiment, we have implemented control measures, and any variances observed in the figures may be attributed to individual variations among the animals. It is also possible that the MFI values measured by flow cytometry exhibit larger variations than a percentage.

Comment 5: Figure 4 H/I - In contrast to NK cells in Fig 4E, F, the KO and WT ILC1s seem to co-cluster largely. Authors should validate differentially expressed genes. How strong is the effect of Blimp1 in ILC1s? Or is Blimp1 a critical TF driving effector differentiation in NK cells, while it has only subtle effects in ILC1 (these may be regulated by Hobit?)? This seems an interesting finding that should at least be discussed. For these types of small differences in ILC1, FACS confirmation analyses should be performed and findings be reevaluated using Cre-expressing controls (see above).

Response 5: We appreciate the suggestion from the reviewer. As request, we analyze the DEGs in liver cNK cells and ILC1s from our scRNA-seq data (revised Supplemental Figure 8, A and B). There only a few valuable DEGs in ILC1s compared to cNK cells. It’s likely that Prdm1 have more essential effect of cNK cells transcriptional program, while it plays more important role in keep the homeostasis of ILC1s population. We have discussed these points to better inform the readers. (page 27; line 554-561):

“Previous studies have identified Hobit and Prdm1 as central regulators instructing tissue-dependent programs and retention of diverse tissue-resident lymphocytes (18, 51, 53). Liver ILC1s required Hobit, but not necessary for cNK cells (6). Expression of Prdm1 was remarkably higher in cNK cells versus ILC1s (18). While in our study, cNK cells and liver ILC1s reduced simultaneously in *Prdm1ΔNcr1* mice, and even more significant in ILC1s. This indicates that while Prdm1 is expressed at lower levels in ILC1s, its role in preserving the quantity of ILC1s may be more crucial. Thus, Prdm1 and Hobit may have parallel program in instructing ILC1s functional development and maturation.”.

We cannot find valuable surface marker to evaluate the change in ILC1s, as most of changes are intracellular markers.

Comment 6: The authors describe and discuss some of Figure 1 and 2 data as if Blimp1 would be involved in alternative NK versus ILC1 fates, but there is no evidence for this.

Response 6: There is no evidence that Prdm1 could alter the fate decision of the progenitor towards liver cNK or ILC1s. Although some studies reported the conversion between cNK cells and ILC1s in special contexts, it was widely accepted that liver cNK cells and ILC1s originated from different progenitors. While we observed changes in the proportions of liver cNK cells and ILC1 in Prdm1 KO mice, we still lack sufficient evidence to support the relative independence of NK and ILC1 development, as well as evidence to indicate that Prdm1 is exclusively responsible for NK and ILC1.

Regarding the changes in NK and ILC1 proportions after Prdm1 KO, we believe that both NK and ILC1 cells require Prdm1 to maintain their populations, with ILC1 possibly requiring it to a greater extent. This is the reason for the altered balance between NK and ILC1 cells following Prdm1 KO. We wish to clarify this point to prevent any misconceptions among readers. To address this, we have added the following content to the discussion section (page 25; line 509-516):

“Furthermore, although both liver NK cells and liver ILC1s require Prdm1 to maintain their quantity, liver ILC1s demonstrate a more pronounced dependency on Prdm1. However, it is currently widely believed that liver NK cells and liver ILC1s originate from different progenitors. It is worth noting that while we observed changes in the NK and ILC1 proportions after *Prdm1* knockout, our data does not support the hypothesis that Prdm1 affects progenitor differentiation decisions, thereby influencing the fate selection of NK and ILC1. Further research may be needed to elucidate how Prdm1 regulates the balance between NK cells and ILC1s.”.

Comment 7: There are several recent studies suggesting a role for Hobit, homologue of Blimp1, in NK cells and in ILC1, and in the control of liver metastases. The authors should discuss similar and unique functions of Hobit and Blimp1, also in the regulation of gene expression patterns, and should refer to these studies.

Response 7: We would like to express our gratitude to the reviewer for your insightful comments, which bring forth a critical perspective. In accordance with the reviewer's suggestion, we have updated our discussion to include the diverse functions guided by Hobit and Prdm1 in regulating the development and function of cNK cells and ILC1s (page 27; line 554-561):

“Previous studies have identified Hobit and Prdm1 as central regulators instructing tissue-dependent programs and retention of diverse tissue-resident lymphocytes (18, 51, 53). Liver ILC1s required Hobit, but not necessary for cNK cells (6). Expression of Prdm1 was remarkably higher in cNK cells versus ILC1s (18). While in our study, cNK cells and liver ILC1s reduced simultaneously in *Prdm1ΔNcr1* mice, and even more significant in ILC1s. This indicates that while Prdm1 is expressed at lower levels in ILC1s, its role in preserving the quantity of ILC1s may be more crucial. Thus, Prdm1 and Hobit may have parallel program in instructing ILC1s functional development and maturation.”.

As shown in Supplemental Figure 8, we analyzed two published scRNA-seq data performed with _Hobit_KO mice and integrated DEGs in cNK cells and ILC1s with our data. We observed overlaps of DEGs in *Prdm1*Δ*Ncr1* and _Hobit_KO between cNK cells and ILC1s, such as *Junb*, *Tcf7*, *Gzmb*, and *Prf1* (Supplemental Figure 8), indicating the similar regulatory network of Prdm1 and Hobit. These data are now described on page 19; lines 386-395:

“We also compared the gene expression patterns between *Prdm1* and *Hobit* (homologue of Blimp1) with two published scRNA-seq data (51, 53). Following the knockout of Hobit, the DEGs were primarily identified within ILC1s. Conversely, after the knockout of Prdm1, a greater number of DEGs were observed in cNK cells. This indicates that Prdm1 likely possesses a broader range of target genes within cNK cells, whereas Hobit appears to have a more pronounced impact on gene expression within ILC1s (Supplemental Figure 8, C-F). There are some overlaps between the downstream transcriptional profile of *Prdm1* and *Hobit* in liver cNK cells and ILC1s (Supplemental Figure 8, G and H), such as *Junb*, *Fosb*, *Tcf7*, *Kit*, *Gzmb*, *Prf1*, and *Cxcr6* was simultaneously upregulated or downregulated in both *Prdm1ΔNcr1* and _Hobit_KO liver cNK cells or ILC1s, indicating the similar regulatory networks of Prdm1 and Hobit.”.

Comment 8: Figure 4: The authors should discuss (and cross-validate) their liver gene expression analyses in the context of published datasets of NK and ILC1, such as the ones by Lopez et al, Friedrich et al, Ducimetiere et al and Yomogida et al.

Response 8: We thank the reviewer for raising this important point. To address this question, we have now analyzed the gene expression of liver cNK cells and ILC1 in two published data mentioned above, also in the context of Hobit-knock. We compared gene expression of different clusters and described in our revised manuscript (page 19; lines 386-395).

“We also compared the gene expression patterns between *Prdm1* and *Hobit* (homologue of Blimp1) with two published scRNA-seq data (51, 53). Following the knockout of Hobit, the DEGs were primarily identified within ILC1s. Conversely, after the knockout of Prdm1, a greater number of DEGs were observed in cNK cells. This indicates that Prdm1 likely possesses a broader range of target genes within cNK cells, whereas Hobit appears to have a more pronounced impact on gene expression within ILC1s (Supplemental Figure 8, C-F). There are some overlaps between the downstream transcriptional profile of *Prdm1* and *Hobit* in liver cNK cells and ILC1s (Supplemental Figure 8, G and H), such as *Junb*, *Fosb*, *Tcf7*, *Kit*, *Gzmb*, *Prf1*, and *Cxcr6* was simultaneously upregulated or downregulated in both *Prdm1ΔNcr1* and _Hobit_KO liver cNK cells or ILC1s, indicating the similar regulatory networks of Prdm1 and Hobit.”.

**Recommendations For The Authors:**
Comment 9: The use of a paired t-test analysis when comparing cells/groups from different mice is not correct. Instead, the authors should consider using e.g. an unpaired t-test and re-test the indicated significance (e.g. Figure 1F, Figure 2H).

Response 9: We thank the reviewer’s comments. As we used littermates for the experiments and they are compared side by side, so the paired t-test analysis is acceptable. We reanalysis the significance in the results of Figure 1F, and Figure 2H using unpaired t-test. The statistics significance of Figure 1F using unpaired t-test was same as using t-test. However, in Figure 2H, the reduced IFN-γ production not reach statistics significance when used un-paired t-test (Supplemental Figure 12B). It may attribute to the variation between different littermates, but the trend is still under the scope of our conclusion. We believe that employing a paired t-test between littermates could be also meaningful. As such, we kept both statistical methodologies to ensure a thorough evaluation.

Comment 10: In several instances, it is unclear whether data are pooled or representative (and if so, of how many analyses). This information needs to be provided for all analyses.

Response 10: We apologize for the lack of details and have now provided the sufficient information in our figure legends.

For example, we delete the number in original histogram to avoid the misunderstanding of the unclear whether data are pooled or representative (e.g. original Figure7 A-C; revised Figure7 A-C). Furthermore, we added the “representative” in figure legends of all flow cytometric plots to better inform readers (e.g. original Figure2, D and F; revised Figure2, B and D).

Comment 11: In the title and abstract authors use "type 1 ILCs" for both NK cells and ILC1, and it is difficult to understand which phenotypes correspond to cNK cells versus ILC1. Most of the analyses clearly separate these two different cell types. I would appreciate a lot being more accurate in the abstract, and describing cNK and ILC1 phenotypes in a clear way.

Response 11: We are really sorry for our inaccurate descriptions. According to Spits et al., (Spits et al., *Nature Reviews Immunology*, 2013) and other related studies, we have now adopted a more appropriate nomenclature as “Conventional NK cells” correspond to “cNK cells”, “Type 1 innate lymphoid cells” to “ILC1s”, and “Group 1 ILC” as the collective name of cNK and ILC1s.

The definition of these cells was described in the introduction (page 4, line 52-53; line58-62):

“Group 1 ILCs consist of cNK cells and ILC1s (1, 2), with distinct developmental trajectories and effect molecules (3).”, “In a state of homeostasis, liver group 1 ILCs (CD45+CD3-NK1.1+NKp46+) can be discriminated into cNK cells and ILC1s by the differential expression of CD49a and CD49b (2): cNK cells are marked by the expression of CD49b, while liver ILC1s exhibit a distinctive positivity for CD49a. Tumor Necrosis Factor Related Apoptosis Inducing Ligand (TRAIL) is also expressed on liver ILC1s, but not on cNK cells (10, 11).”.

We also describe cNK and ILC1 phenotypes in our scRNA-seq data, as shown in page 13; line 259-261:

“cNK cells expressed high levels of *Itga2* (CD49b) and Eomes, while ILC1s had high levels expression of *Itga1* (CD49a) and *Tnfsf10* (Supplemental Figure 5, F and G).”.

Comment 12: In the abstract authors state "The present study unveiled a novel regulatory mechanism of Prdm1 in liver Type 1 ILCs, showing promising potential for developing innovative immune therapy strategies against liver cancer." - maybe authors should discuss how their findings could be used for therapeutic approaches?

Response 12: We appreciate comments from the reviewer. As there hasn't been a clear consensus on the role of Prdm1 in NK cells prior to this, some studies have suggested that Prdm1 can inhibit cytokine secretion by NK cells. Particularly, Kallies et al. in their 2011 article in *Blood* found that Prdm1 might suppress NK cell anti-tumor activity. Hence, there hasn't been any immunotherapy targeting Prdm1 in NK cells for cancer treatment. Our research demonstrates the enhancing role of Prdm1 in NK cell anti-tumor activity, providing theoretical support for NK cell therapy targeting Prdm1.

We added the following content to the discussion section (page 29; line 605-609):

“Further research may provide deeper insight into the role of PRDM1 in the anti-tumor function of human NK cells, enabling a more direct investigation of its application in cancer therapies. Given its important role in preserving liver cNK cells and ILC1s functional heterogeneity, enhancing Prdm1 function in human NK cells could potentially be a strategy to promote NK cell-based immunotherapy for cancer.”.

Comment 13: The authors should explain or interpret their data a bit more (e.g.) what is the consequence of GSEA enriched in negative regulation of Il6 production? (Fig. 3D) do NK cells produce Il6 (Figure 3)? What's the impact of Il17 signaling in NK/ILC1 (Figure 5). Do the authors suggest JunB-driven metabolic reprogramming (Suppl. Fig 6D-F?).

Response 13: We appreciate comments from the reviewer. The question of IL-6 production in NK cell also raised by another reviewer. We have checked the GSEA results, and found no valuable genes in IL-6 production in NK cells. According to the suggestions of another reviewer (Response to Reviewer 2 Comment, Comment 14), it may be prudent to omit this figure.

IL-17 signaling indicated the plasticity of ILC1s, that may be originated from the differentiation of ILC3, we added more discussion of this part (page 17; line 341-344).

“Several ILC3 signature genes, such as *Rora*, *Tmem176a,* and *Tmem176b* (45), highly expressed in this cluster (Supplemental Figure 7D). Considering the close relationship between IL-17 mediated immunity response and ILC3 (1, 46), it is plausible that _Il7r_hi ILC1 cluster may be attributed, at least in part, to potential plasticity between ILC1 and ILC3 subsets.”.

The decreased mitochondrial function may have more relevance to NK cell exhaustion in tumors. Our data suggest that the elevated expression of JunB in NK cells may predispose them to exhaustion. Currently, our hypothesis regarding the promotion of NK cell exhaustion by high JunB expression is based on the observed correlation between JunB expression levels and exhaustion phenotypes (at the gene expression and IFN-γ secretion levels) and the findings in reference 67 (Lynn et al., *Nature*, 2019), where JunB was found to promote T cell exhaustion. However, we have not demonstrated causation between high JunB expression and exhaustion in NK cells. We propose that in NK cells, especially mature NK cells, excessive JunB expression may make them more sensitive to exhaustion inducers. Nevertheless, further research is needed to confirm this. To clarify this, we added the following content in the discussion section (page 26; line 537-543):

“While our current data is not sufficient to definitively classify these cells as exhausted NK cells, it supports that a subpopulation, referred to *Junbhi* cluster, demonstrates an exhaustion-like phenotype.

The significant increase in this cell population following *Prdm1* knockout in NK cells may potentially be one of the reasons why *Prdm1ΔNcr1* mice lose their tumor-killing capacity. Whether the excessive expression of JunB in NK cells is also a contributing factor to their exhaustion, similar to T cells(65), requires further investigation.”.

Comment 14: Ref 25 and Ref 57 are the same publication?

Response 14: We are really sorry for our careless mistakes. We have checked all the reference and corrected the wrong format.

Comment 15: Figure 1, E - The method description of RT-PCR is missing. I apologize if I have overlooked this information.

Response 15: We have now added the description of RT-PCR in our revised method section (page 31; line 638-644):

“RNA was extracted from FACS-sorted NK cells or splenocytes using RNASimple Total RNA Kit (TIANGEN Biotech, 4992858) and subsequently reverse transcribed to cDNA with SuperScript VILO Master Mix (Thermo Fisher Scientific, 11755050) according to manufacturer’s instructions. qPCR was performed with SYBR Green Mix (Thermo Fisher Scientific, A25742) and CFX Opus 96 Real-Time PCR System (Bio-Rad). The relative mRNA expression level was calculated using 2-ddCt method. Primer sequences: Prdm1: 5’-CAGAAACACTACTTGGTACA-3’; 5’-GATTGCTTGTGCTGCTAA-3’.”

Comment 16: Figure 1, F - The NKp46+CD3- gate for the liver seems to cut the population, not all cells are included.

Response 16: We appreciate the review’s comment and apologize for our carelessness. We expend our data with more samples and reanalyzed them with a more convincing gating strategy. We now update our figures (revised Figure 1G; revised Supplemental Figure 2A). Several changes have occurred in the data and conclusions, and we have accordingly revised these contents in our manuscript.

The original text is:

“Proportion and absolute number of cNK cells in blood, bone marrow, lung, liver, spleen, and lymph nodes were analyzed by flow cytometry. Compared with *Prdm1+/+* mice, the percentage of cNK cells (CD3-NK1.1+NKp46+) among lymphocytes was decreased in all of these tissues except bone marrow and lymph nodes (Figure 1F; Supplemental Figure 2A). However, no significant difference was observed in the percentage of cNK cells among bone marrow-derived lymphocytes between *Prdm1ΔNcr1* and *Prdm1+/+* mice. The absolute number of cNK cells in blood, lung, liver, and spleen also decreased in *Prdm1ΔNcr1* mice (Figure 1F; Supplemental Figure 2A). Only a slight decrease in the number of cNK cells was observed in the lymph nodes of *Prdm1ΔNcr1* mice, which did not reach statistical significance either (Supplemental Figure 2A). In contrast, the absolute number of cNK cells in *Prdm1fl/fl* mice bone marrow is moderately higher than *Prdm1ΔNcr1* mice (Figure 1F).”

The revised text is (page 8; line 142-146):

“Proportion and absolute number of cNK cells in blood, bone marrow, lung, liver, spleen, and lymph nodes were analyzed by flow cytometry. Compared with *Prdm1+/+* mice, the percentage and absolute number of NK cells (CD45+CD3-NK1.1+NKp46+) among lymphocytes was decreased in all of these tissues, whereas increased number of NK cells were observed in bone marrow (Figure 1G; Supplemental Figure 2A).”

Comment 17: Figure 1, The y-axis labeling of lung CD3-NKp46+ cells (x10^3) is not correct.

Response 17: We are really sorry for our carelessness. We now check the labels and make sure they are correct.

Comment 18: Figure 1, The statistical significance of absolute numbers of NKp46+ cells in the bone marrow should be reviewed.

Response 18: We expend our data with more samples and reanalyzed them with a more convincing gating strategy. We observed significant increase of bone marrow NK cells quantity in our updated data. These changes are now described in our revised manuscript.

The original text is:

“However, no significant difference was observed in the percentage of cNK cells among bone marrow-derived lymphocytes between *Prdm1ΔNcr1* and *Prdm1+/+* mice”, “In contrast, the absolute number of cNK cells in *Prdm1fl/fl* mice bone marrow is moderately higher than *Prdm1ΔNcr1* mice (Figure 1F).”

The revised text is (page 8; line 142-146):

“Proportion and absolute number of cNK cells in blood, bone marrow, lung, liver, spleen, and lymph nodes were analyzed by flow cytometry. Compared with *Prdm1+/+* mice, the percentage and absolute number of NK cells (CD45+CD3-NK1.1+NKp46+) among lymphocytes was decreased in all of these tissues, whereas increased number of NK cells were observed in bone marrow (Figure 1G; Supplemental Figure 2A).”

Comment 19: Figure 1, G - CD27 and CD11b are used to define maturation stages within NK cells. Here the authors are analyzing group 1 ILC instead (containing both NK cells and ILC1, especially in the liver). It would be better to pre-gate on Eomes+ or CD49b+ NK cells for this analysis.

Response 19: We apologize for the lack of details in this analysis. We have pre-gate CD49b+ NK cells for the maturation stages analysis. We have now added this statement in our revised manuscript and figure legend (page 8; line 149-151)

“The maturation of cNK cells (gated by CD45+CD3-NK1.1+NKp46+CD49b+) from blood, bone marrow, lung, liver, spleen, and lymph nodes were assessed, based on the expression of CD11b and CD27.”.

Comment 20: Supplementary Figure 1, A - The NKp46+CD3- gate seems to cut the population, not all cells are included. y-axis labeling of spleen CD3-NKp46+ cells (%) is not correct.

Response 20: Thanks, we have corrected these errors and shown in our revised supplementary Figure 2A.

Comment 21: Figure 2, D-G - Did the authors analyse the ILC1/NK compartment of the tumor? What is the abundance and phenotype of these cells dependent on Prdm1 expression? Proper Crecontrols should be used (see above).

Response 21: We appreciate the suggestions from the reviewer. As request, we have now added the analysis of cNK/ILC1s population in the context of tumor. The proportion changes of cNK cells and ILC1s in *Prdm1*Δ*Ncr1* mice was similar with the no tumor-burden condition, while the number of both cNK cells and ILC1s decreased in tumor bearing liver (revised Figure 7D). These contents have been updated in our revised manuscript (page 23; line 479-481):

“The proportion changes of cNK cells and ILC1s in *Prdm1ΔNcr1* mice was similar with the no tumorburden condition, while the number of both cNK cells and ILC1s have significant decreased in tumor-bearing liver (Figure 7D).”.

The reason why we did not use Cre-controls was described in comment 1.

Comment 22: Figure 2, H - Prdm1-deficient NK and ILC1 produce less Ifng in response to in vitro stimulations with Il-12 and /or Il-18, and bulk Seq analysis (Fig 3F) shows reduced Il12rb2 expression. Does the expression of cytokine receptors correlate with the maturation of NK cells? This could be analyzed from the single-cell RNA-seq dataset. The statistical significance of %Ifng after Il12/Il18 stimulation should be revisited (see above).

Response 22: We thank the reviewer for the suggestions. To address this question, we explored the expression of IL-12 and IL-18 receptors in cNK and ILC1 clusters. Within cNK clusters, *Il12rb2*, *Il18r1* and *Il18rap* was highly expressed in *Prf1hi* and *Cxcr3hi* cNK clusters (revised Supplemental Figure 6H), indicating the IL-18 receptor expression correlated with the NK cell maturation. While in ILC1, these receptors mostly expressed on _Il7r_hi and _Gzmb_hi ILC1 clusters (revised Supplemental Figure 7C). Significant decreased of *Il18r1* expression in *Prdm1*Δ*Ncr1* cNK cells and ILC1s may associated with the impaired ability to produce IFN-γ. We now added this analysis (page 18; line 364-368):

“Within cNK cells, *Il12rb2*, *Il18r1* and *Il18rap* was highly expressed in *Prf1hi* and *Cxcr3hi* cNK clusters (Supplemental Figure 6I), indicating the IL-18 receptor expression correlated with the NK cell maturation. While in ILC1, these receptors mostly expressed on _Il7r_hi and _Gzmb_hi ILC1 clusters (Supplemental Figure 7D). Significant decreased of *Il18r1* expression in *Prdm1ΔNcr1* cNK cells and ILC1s may associated with the impaired ability to produce IFN-γ.”.

The un-paired t test of IFN-γ production was displayed in revised supplemental Figure 12 B. Difference in IFN-γ production was found to be not significant when analyzed using an unpaired ttest in original Figure 2 H. However, significance was observed in tumor-bearing liver cNK cells and ILC1s, specifically under the context of IL-12/IL-18 stimulation, as depicted in the original Figure 7E using an unpaired t-test. These variations may be attributed to differences among different littermates. Despite these variations, the trend remains consistent with our overall conclusions. We believe that employing a paired t-test between littermates could be also meaningful. As such, we kept both statistical methodologies to ensure a thorough evaluation.

Comment 23: Figure 3, A-E - For bulk sequencing analysis, splenic CD3-NK1.1+NKp46+ were isolated. This population also contains ILC1 in the spleen (e.g. Flommersfeld et al.), although much less abundant compared to NK cells, and compared to the liver compartment. However, have the authors tested the abundance of splenic ILC1 in Prdm1-deficient mice, which may impact the gene expression data? In line with this the detection of altered Cxcr6 expression in Figure F, which is usually expressed by ILC1 rather than NK cells, may indicate an alteration in ILC1 numbers. The authors should validate the altered expression of CXCR6, Itga1, and Cx3cr1 on NK cells by flow cytometry.

Response 23: We cited the work of Flommersfeld et al. into our manuscript and have expanded our Results section to include the following information (page 19; line 377-385):

“Previous research found that spleen NK cells could be divided into three distinct groups based on their expression levels of CD27, CD62L, CD49a, and CD49b (52). CD27+CD62L- NK cells have remarkable high expression of Batf3, while it was only barely expressed in CD27+CD62L+ and CD27-CD62L+ NK cells (52). Based the sequencing data published by Flommersfeld et al., (GSE180978), a notable negative correlation was observed between the expression levels of Prdm1 and Batf3 (Supplemental Figure 8I). On top of that, our findings unveiled the negative regulatory influence of Prdm1 on Batf3 within both spleen and liver NK cells. This discovery highlights a potential upstream mechanism that may influence the hemostasis of the spleen NK cell subpopulations through Batf3.”.

We validated the expression of CD49a (Itga1) and CX3CR1 in liver cNK cells and ILC1s in our revised manuscript, which is described in our revised manuscript (page 9; line 170-174, page 14; line 231-233):

“Increased CD49a expression was also observed in *Prdm1ΔNcr1* liver ILC1s, while it showed decreased expression in NKp46+ cells in the liver, bone marrow, and lymph nodes (Supplemental Figure 2, F and G).”, “The percentage of CX3CR1+ cNK cells was significantly decreased in multiple tissues of *Prdm1*Δ*Ncr1* mice, while the proportion of CX3CR1+ ILC1 was increased in the liver (Figure 3F).”

Comment 24: Figure 3, F - Tnfsf26: which gene is this? is this a typo? Is a function of this gene in NK cells reported? Altered Batf3 expression suggests an impact on ILC1-like NK cells (Flommersfeld et al).

Response 24: We are very sorry for our mistakes. We have removed Tnfrsf26 from the heatmap.

Comment 25: Figure 3, G-J refer to Kallies data?!

Response 25: Kallies‘s data has mentioned the reduced GzmB expression in Blimp1gfp/gfp mice. However, compared with Kallies’s study, we further analyzed the GzmB and Perforin expression in different mature stages of NK cells. Reduced GzmB expression not only due to the less mature phenotype in *Prdm1*-deficient NK cells, highlighting the role of Prdm1 in regulating NK cell function. So, we added these contents in the revised manuscript (page 12; line 233-242):

“Lower GZMB and PRF1 production was observed in *Prdm1*-deficient splenic cNK cells, liver cNK cells and ILC1s (Figure 3, H-K; Supplemental Figure 4, A-I). Notably, the proportion of GZMB+ and PRF1+ cNK cells was decreased among almost all of the maturation stages of cNK cells (Figure 3, J and K). The relative mean fluorescent intensities (MFIs) of GZMB and PRF1 consistently show a reduction across all developmental stages in *PrdmΔNcr1* NK cells (Supplemental Figure 4, H and I). Yet, no statistical difference of PRF1 was found within the CD11b-CD27+ and CD11b+CD27+ subsets, likely due to the relatively lower perforin levels in these populations (Supplemental Figure 4I). These findings suggest that Prdm1 may directly influence cytotoxic molecule in NK cells, rather than impacting their anti-tumor abilities solely by affecting the maturation phenotype of Prdm1-deficient NK cells.”

In Discussion section (Kallies’s work is cited here in revised manuscript) (page 24; line 500-502):

“Our results not only confirmed a decrease in cytotoxic molecules in *Prdm1*-deficient NK cells (31) but also showed that the reduction in Gzmb and perforin is not solely attributable to the diminished maturation of these cells.”

Comment 26: Figure 3, G, I - How do the authors explain the high variability of GzmB and Prf1 in Prdm1+/+ cells? 2 samples have comparable values to Prdm1-deficient cells.

Response 26: This may be due to the inherent differences in MFI among different samples. In the revised version, we have added data on percentages, which exhibit much less variability (Figure 3, H and I). The MFIs of GZMB and PRF1 are moved to supplemental Figure 4 E and F.

Comment 27: Did the authors test the mice for potential germline recombination of the floxed allele, which has been suggested as a potential problem of Ncr1cre?

Response 27: We appreciate the insightful comments provided by the reviewer, and this is a really good question. In *Prdm1fl/fl* mice, germline recombination typically results in a systemic knockout of *Prdm1*, which can lead to embryonic lethality. Given that mice were successfully born in the current study, it is almost unlikely that germline recombination of Prdm1 occurred due to leaky expression of Cre.

To confirm this issue, we isolated splenocytes and assessed Prdm1 expression using qPCR. We observed no significant difference in Prdm1 expression between splenocytes from *Prdm1+/+* and *Prdm1ΔNcr1* mice (revised Figure 1F). This also indicated that germline recombination issues are unlikely to be present in the *Prdm1ΔNcr1* mice.

Comment 28: Histograms do not show MFI

Response 28: We appreciate the comments provided by the reviewer. The MFI value was omitted.

Comment 29: Supplementary Figure 4, B - FACS plot labelling: Typo, Histograms do not show MFI.

Response 29: We sincerely thank the reviewer for careful reading. The typo in this figure was corrected. The MFI is omitted.

Comment 30: Figure 4, A - What are the cells in the red cluster in the middle of the UMAP, do they belong to B cells? Why do they cluster so separately? It is interesting, but also surprising that NK and ILC1 cluster map so far apart from each other (rather with CD8 or B cells? or NKT cells) - do the authors have any comments?

Response 30: We sincerely apologize for the mistakes in labeling a group of cells in our previous analysis. Upon a thorough re-evaluation, we have corrected the labels of several cell clusters that were previously misidentified. The revised heatmap (revised Supplemental Figure 5C) represents the marker genes for each cluster. Additionally, in our updated analysis (revised Figure 4A), we have included clusters for Epithelial cells, CD4+ T cells, NKT cells, and Kupffer cells. Please note, the red cluster identified in the center of the original heatmap corresponds to the CD4+ T cells.

We checked the markers of cNK cell and ILC1 clusters and confirmed they are labeled correctly, as *Ncr1* and *Klrb1c* (NK1.1) was highly expressed in these clusters compared to others (revised Supplemental Figures 5E).

Comment 31: Does Junb expression correlate with the maturation stages of NK cells?

Response 31: Our previous research indicated that during the maturation process of NK cells, there was a decrease in the expression levels of Junb (negative correlation), whereas there was an increase in the expression levels of Prdm1 (Wang et al., *J Clin Invest*, 2018; Supplemental Figure 5c and Supplemental Figure 11).

Comment 32: The authors may consider validating their scRNA-seq data (e.g. by FACS analysis for highlighted markers, eg. cKit, Tcf7, Gzma, Cxcr3).

Response 32: We appreciate the suggestion from the reviewer. We validated several marker genes, including Gzmb, Prf1, and Cx3cr1 by FACS, as shown in the revised Figure 3 F-K. Currently, FACS cannot distinguish liver NK cells into as many distinct clusters as can be achieved through scRNAseq analysis. However, we expect that as technology progresses, we will be able to enhance our validation of the scRNA-seq data.

Comment 33: It is a bit unclear to me why authors refer to Cxcr3hi NK cells as tissue-resident. This is based on Cxcr3 and Ccr2 expression. To make this statement, a much more detailed analysis would be required. How are CD69, CD49a, or CXCR6 expression of these cells?

Response 34: We appreciate the suggestion from the reviewer. The primary reason for classifying this specific cluster of NK cells as tissue-resident is derived from the differential expression genes (DEGs) and Gene Ontology (GO) analysis, which demonstrate significant chemokine receptor activity within this cluster.

To make this statement more clearly, we check the expression of the above markers, but only Cd69 had expression in cNK clusters, which was highly expressed in _Junb_hi and _Cxcr3_hi cNK cells (revised Supplemental Figure 6D). We also used top30 DEGs in ILC1s versus cNK to calculate the module score in all cNK clusters, as _Cxcr3_hi cNK had highest score among these clusters (revised Supplemental Figure 6D). This part has been updated in our manuscript (page 15; line 298-308):

“Expression of tissue-resident markers Cd69 was also highly expressed in this clusters (Supplemental Figure 6D). The enrichment of chemokine receptors in the genes upregulated in the _Cxcr3_hi cluster implying a greater likelihood of this cluster being tissue-resident compared with other cNK cell clusters (Figure 4H). To further confirmed tissue-resident properties of this clusters, we calculated the module score based on top30 DEGs in ILC1 versus cNK clusters, including *Cxcr6*, *Itga1*, *Cd160*, *Cd226*, etc. _Cxcr3_hi cNK clusters have the highest score among all cNK clusters (Supplemental Figure 6H), indicating the similarity with liver ILC1s. In the tumor microenvironment, reports indicated that NK cells could transform into ILC1s (25). If this conversion of cNK cells into ILC1s also occurred under normal physiological conditions, then _Cxcr3_hi cNK cell cluster might be the most susceptible to such transformation.”

Comment 35: The authors suggest that Prdm1 regulates chemokine receptor expression. An alternative explanation could be that this is an indirect effect of altering the abundance of NK cell subsets.

Response 35: We are sorry for lacking the details in these figures. The input cell number of each genotype has now been added in following figure legends.

Figure 4F, “Proportions of cNK cells among total cNK cells (left; 211 cells in *Prdm1+/+*, and 141 cells in *Prdm1ΔNcr1*) and within clusters (right).”; Figure 5C, “Proportions of ILC1s among total ILC1s in different genotypes (left; 114 cells in *Prdm1+/+*, and 63 cells in *Prdm1ΔNcr1*) and within each cluster (right).”; Figure 6C, “Proportions of MDMs and KCs among total macrophages in different genotypes (510 cells in *Prdm1+/+*, and 624 cells in *Prdm1ΔNcr1*).”

To minimize the effects of discrepancies in input numbers between samples with different genotypes, we represented the relative proportions of each cluster within its specific genotype (e.g. Supplemental Figure 6B; Supplemental Figure 7B; Supplemental Figure 9B).

Comment 36: Supplementary Figures 6 and 7, A - The formatting of gene annotations does not fit the heat maps (the gene names on the last rows are missing).

Response 36: We apologize for our careless mistakes. We have now addressed these mistakes.

Comment 37: Supplementary Figures 6 and 7, What is the consequence of compromised mitochondrial function? Increase apoptosis?

Response 37: In our experiments, we did not find that Prdm1 has an effect on the apoptosis of NK cells. Conversely, previous studies have found that Prdm1 might inhibit the proliferation of NK cells (C. Kucuk, et. al., PNAS, 2011). We acknowledge that there is ongoing debate regarding the precise definition of NK cell exhaustion. In our experiments, no changes were detected in the expression levels of surface markers (TIGIT) associated with exhaustion on NK cells following the knockout of Prdm1. However, we did note a significant reduction in the cytokine secretion capacity and tumor control efficacy of NK cells after Prdm1 knockout. We prefer to say that the consequence of compromised mitochondrial function might be increased exhaustion. As we mentioned in discussion part (line 482-483), mitochondrial fragmentation has been confirmed to be closely associated with NK cell exhaustion in tumor (Zheng et al. *Nature immunology*, 2019). Although the evidence to define the exhausted NK cells in *Prdm1*Δ*Ncr1* was not sufficient, our data may support the compromised mitochondrial functions, at least in part, associated with the exhausted phenotype of *Prdm1*Δ*Ncr1* NK cells in cancer.

We have discussed these points in our revised manuscript (page 26; line 529-543):

“Mitochondria are pivotal organelles crucial for cellular metabolism. Disruptions in mitochondrial function have been linked to T Cell exhaustion, attributed to glycolytic reprogramming (66). Similarly, mitochondrial fragmentation has been closely associated with NK cell exhaustion (67).

However, the concept of NK cell exhaustion isn't as firmly established as it is for T cells. Exhausted NK cells should primarily exhibit diminished functions. This is characterized by a diminished ability to destroy tumor cells, a reduced capability to activate other components of the immune system, and compromised proliferation and survival rates. Additionally, this reduced functionality is associated with a decline in the expression of molecules responsible for cytotoxic activity, lower production of IFN-γ, and metabolic disturbances that may arise from mitochondrial dysfunction. While our current data is not sufficient to definitively classify these cells as exhausted NK cells, it supports that a subpopulation, referred to _Junb_hi cluster, demonstrates an exhaustion-like phenotype. The significant increase in this cell population following *Prdm1* knockout in NK cells may potentially be one of the reasons why *Prdm1ΔNcr1* mice lose their tumor-killing capacity. Whether the excessive expression of JunB in NK cells is also a contributing factor to their exhaustion, similar to T cells(65), requires further investigation.”.

Comment 38: Figure 5, Describing the scRNA Seq data, the authors are switching a lot between Figure 4 and Figure 5. Maybe a reorganization of the Figures (Figure 4: NK cell; Figure 5: ILC1) could help.

Response 38: We appreciate the reviewer’s suggestion. We have now reorganized the Figure 4 and Figure 5.

Comment 39: Figure 5, We suggest naming one of the ILC1 clusters "Gzmbhi" to keep it consistent with the FACS data.

Response 39: We agree with this excellent suggestion and have now renaming the “Gzmahi” ILC1 cluster as “Gzmbhi” ILC1 cluster.

Comment 40: Figure 5, C - How was the JunB score derived (which genes were used)?

Response 40: The JunB score was calculated based on the expression of marker genes in _Junb_hi cNK clusters (DEGs in _Junb_hi cNK cluster compared to other clusters, as shown in revised Supplemental figure 6A). The score was calculated using “AddModuleScore” R package.

Comment 41: Figure 5, G, I - The authors highlight Il17 signaling pathway, what is the impact of Il17 on NK/ILC1? Did the authors check for ILC3 (Rorc expression) within the ILC1 cluster?

Response 41: The enrichment of IL-17 signaling pathway in _Il7r_hi ILC1 indicated that this cluster encompass ILC1s originate from the conversion of Rorγt+ ILC3s. Although the Rorc expression was undetectable in all ILC1 clusters, we found several ILC3 marker genes highly expressed in this clusters (e.g. Rora, Tmem176a, Tmem176b) according to the ILC3 transcriptomes (Robinette et al., *Nature Immunology*, 2015).

We have added these contents in our revised manuscript (page 17; line 341-344):

“Several ILC3 signature genes, such as *Rora*, *Tmem176a,* and *Tmem176b* (45), highly expressed in this cluster (Supplemental Figure 7D). Considering the close relationship between IL-17 mediated immunity response and ILC3 (1, 46), it is plausible that _Il7r_hi ILC1 cluster may be attributed, at least in part, to potential plasticity between ILC1 and ILC3 subsets.”.

Comment 42: Figure 5, The authors detect more Ly49E+ cytotoxic ILC1 in Prdm1fl Ncr1cre mice.How does this observation fit to the reduced cytotoxicity of NK cells?

Response 42: The proportion of _Klra_hi ILC1 was increased, while the _Gzmb_hi ILC1 was decreased in *Prdm1*ΔNcr1 mice. Moreover, total number of three ILC1 cluster was reduced in *Prdm1*ΔNcr1 mice.

Comment 43: Line 350/351: Citation required.

Response 43: We added the respective reference. (reference 55 and 56).

Comment 44: Figure 6, The Cell-chat analysis provides interesting suggestions, but none are experimentally addressed. It is also difficult to evaluate these analyses: are any of the Mac subsets altered in frequency or phenotype in either genotype? This could be analyzed from the single-cell data in Fig 4. At the very least, flow cytometric validation of predicted shifts in the Mac compartment should be confirmed.

Response 44: We gratefully thanks for these valuable suggestions. As requested, we analyzed macrophages and validated some of the scRNA-seq data by flow cytometry. We have re-written this part with the analysis of altered proportion of two macrophage clusters (Kupffer cells and Monocyte-derived macrophages) (page 20-21; line 399-436):

“The scRNA sequencing analysis identified two well-established subpopulations of liver macrophages: the resident Kupffer Cells (KCs) and the Monocyte-Derived Macrophages (MDMs) (Figure 6, A-C; Supplemental Figure 9A). When comparing the total proportion of macrophages within the immune cell population of the liver between WT and *Prdm1ΔNcr1* mice, there is an increase in *Prdm1ΔNcr1* mice (Figure 6C). To confirm these findings, we utilized flow cytometry to define macrophages, including both KCs and MDMs, gating by CD45+Ly6G-F4/80+CD11b+ (Figure 6D).

Our analysis showed that, following the deletion of *Prdm1* in Group 1 ILCs, there is a significant increase in both the proportion and number of macrophages in the liver (Figure 6D).

According to the transcriptional profile, liver macrophages further clustered and were labeled as “_Ly6c2_hi”; “_Cxcl2_hi”; “_Ear2_hi” MDMs, and “_Mrc1_hi”; “_C1q_hi” KCs (Figure 6A, Supplemental Figure 9, A-E). Increased proportion of MDMs and KCs was observed in *Prdm1ΔNcr1* cells (Supplemental Figure 9B). Within MDMs clusters, _Ly6c2_hi MDMs mainly compose of *Prdm1+/+* cells, while *Prdm1ΔNcr1* cells concentrated in _Cxcl2_hi cluster (Figure 6C). The scRNA-seq data reveal that following Prdm1 knockout in NKp46+ cells, there is a decrease in the proportion of KCs within the macrophage population, while the proportion of MDMs increases (Figure 6D). CX3CR1, a chemokine receptor, is extensively utilized to distinguish KCs and MDMs within macrophages. Cells expressing CX3CR1 are identified as MDMs, whereas those without CX3CR1 expression are categorized as KCs (56). Employing flow cytometry and leveraging CX3CR1 expression, we assessed the ratios of KCs and MDMs. However, diverging from the scRNA-seq findings, flow cytometry indicates that post-Prdm1 knockout in group 1 ILCs, there is a minor increase in the proportion of KCs within the total liver macrophages, and a decrease in the proportion of MDMs (Figure 6D; Supplemental Figure 9B). This discrepancy could stem from the different bases of classification: scRNA-seq defines KCs based on gene expression profiles, whereas flow cytometry differentiates between KCs and MDMs using the single surface marker, CX3CR1. Analysis of the macrophage subsets identified by scRNA-seq reveals that, while MDM clusters generally show high CX3CR1 expression, there exists a subset within MDMs, labeled *Mrc1hi*, that also exhibits high levels of CX3CR1 (Supplemental Figure 9C). Consequently, if flow cytometry solely employs CX3CR1 for differentiating between KCs and MDMs, it could result in disparities when compared to scRNA-seq outcomes. Both KCs and MDMs has significantly increased in *Prdm1ΔNcr1* mice, which was consist with the scRNA-seq data (Supplemental Figure 9, B and F). Despite the decrease in the proportion of *Ly6c2hi* MDMs in *Prdm1ΔNcr1* mice, the expression levels of *Ly6c2* exhibited minimal variation between WT and *Prdm1ΔNcr1* mice (Supplemental Figure 9D). Intriguingly, within certain cellular subsets, notably the *Ear2hi* cluster, the *Ly6c2* expression levels in KO mice were found to be higher than those in WT mice. Additionally, we employed flow cytometry to examine Ly6C expression within the macrophages. Similar with the scRNA-seq findings, there were no notable differences in Ly6C expression levels between WT and KO mice (Figure 6E; Supplemental Figure 9G).”.

The changes of the macrophage compartment indicated the potential influence of functional NK cells to macrophages. We have revised these parts in our results and discussion (line 590-601). However, to address more analysis on macrophage is worthy but would go beyond the scope of this manuscript, which will be a direction of our further work.

Comment 45: Figure 6, C1qhi Mac only are few cells/events, and interactions (or cells?) seem to be gone in the Prdm1-floxed mice. Is that true? Does it make sense to perform cell-chat analysis on so few cells?

Response 45: We have now added KCs to the cell-chat analysis, and this cluster was belonged to C1qhi KCs. We have revised the analysis of corresponding parts in our manuscript (page 20-21; line 408-428):

“According to the transcriptional profile, liver macrophages further clustered and were labeled as “_Ly6c2_hi”; “_Cxcl2_hi”; “_Ear2_hi” MDMs, and “_Mrc1_hi”; “_C1q_hi” KCs (Figure 6A, Supplemental Figure 9, A-E). Increased proportion of MDMs and KCs was observed in *Prdm1ΔNcr1* cells (Supplemental Figure 9B). Within MDMs clusters, _Ly6c2_hi MDMs mainly compose of *Prdm1+/+* cells, while *Prdm1ΔNcr1* cells concentrated in _Cxcl2_hi cluster (Figure 6C). The scRNA-seq data reveal that following Prdm1 knockout in NKp46+ cells, there is a decrease in the proportion of KCs within the macrophage population, while the proportion of MDMs increases (Figure 6D). CX3CR1, a chemokine receptor, is extensively utilized to distinguish KCs and MDMs within macrophages. Cells expressing CX3CR1 are identified as MDMs, whereas those without CX3CR1 expression are categorized as KCs (56). Employing flow cytometry and leveraging CX3CR1 expression, we assessed the ratios of KCs and MDMs. However, diverging from the scRNA-seq findings, flow cytometry indicates that post-Prdm1 knockout in group 1 ILCs, there is a minor increase in the proportion of KCs within the total liver macrophages, and a decrease in the proportion of MDMs (Figure 6D; Supplemental Figure 9B). This discrepancy could stem from the different bases of classification: scRNA-seq defines KCs based on gene expression profiles, whereas flow cytometry differentiates between KCs and MDMs using the single surface marker, CX3CR1. Analysis of the macrophage subsets identified by scRNA-seq reveals that, while MDM clusters generally show high CX3CR1 expression, there exists a subset within MDMs, labeled *Mrc1hi*, that also exhibits high levels of CX3CR1 (Supplemental Figure 9C). Consequently, if flow cytometry solely employs CX3CR1 for differentiating between KCs and MDMs, it could result in disparities when compared to scRNA-seq outcomes.”.

Comment 46: Figure 6, C - Here the interactions of both Mac+ILC1 and Mac+NK are shown together. It would be interesting to separate this analysis (also Suppl. Fig 9A-B) into comparisons of Mac+ILC1 vs Mac1+NK from WT or Prdm1fl Ncr1 mice.

Response 46: As request, we re-analyzed this part in each genotype, which was showed in the Supplemental Figure 10. These data have now been described in (page 22; line 445-447).

“The reduction of interaction mostly occurred in the cross-talk of ILC1-MDM and ILC1-KC, whereas no difference was observed in cNK-MDM and cNK-KC interaction (Supplemental Figure 10, A-H)”

Comment 47: Supplementary Figure 9, A, B - Is this analysis using WT and Prdm1fl Ncr1cre dataset together?

Response 47: Yes, we used WT and *Prdm1*Δ*Ncr1* data together. As the request above, we separate this analysis from WT or *Prdm1*Δ*Ncr1* Ncr1 mice. These data have now been described in (page 22; line 445-460):

“The reduction of interaction mostly occurred in the cross-talk of ILC1-MDM and ILC1-KC, whereas no difference was observed in cNK-MDM and cNK-KC interaction (Supplemental Figure 10, A-H). A reduction in the interaction of ligand-receptor, such as Mif-CD74, Cxcl16-Cxcr6, and Cxcl10-Cxcr3 was observed in *Prdm1ΔNcr1* mice compared to *Prdm1+/+* mice (Supplemental Figure 11). Compared to *Prdm1+/+* mice, the information flow of CXCL and MIF pathways significantly decreased in *Prdm1ΔNcr1* mice (Figure 6, H and I; Supplemental Figure 10, B, D, F, and H). These pathways play a crucial role in facilitating macrophage migration. The CXCL signaling was sent from _Ly6c2_hi _Cxcl2_hi MDMs and _C1q_hi KC, targeting all ILC1 clusters and _Cxcr3_hi cNK cell clusters (Figure 6J). Of note, although the population of _Cxcl2_hi macrophage primarily comprised cells from *Prdm1ΔNcr1* mice, the interaction within the CXCL pathway between macrophages and group 1 ILCs was obviously less than *Prdm1+/+* sample (Figure 6J). These changes could be linked to a decreased population of ILC1s and _Cxcr3_hi cNK cell cluster in *Prdm1ΔNcr1* mice, implying that the homeostasis of _Cxcl2_hi macrophages required sufficient signals from cNK cells and ILC1s. The impaired CXCLCXCR interactions might subsequently lead to reduced recruitment and activation of group 1 ILCs and macrophages within the tumor microenvironment.”.

Comment 48: Figure 7, A-C -What is the consequence/interpretation of reduced Mitotracker staining? Any metabolic assays performed? The definition of NK cell "exhaustion" is unclear, is reduced IFNg enough for that? Is the concept of NK cell exhaustion clearly established? Only shortly touched upon in the discussion, the rationale for suggesting an exhausted phenotype, should be explained.

Response 48: MitoTracker was used to assess the mitochondrial mass. The reduced staining indicated compromised mitochondria function, which associated with mitochondrial fragmentation.

We believe that the exhaustion of NK cells is not as well-established a concept as it is for T cells. The purpose of detecting mitochondria in this study is to provide evidence for the relationship between Prdm1 and the exhaustion of NK cells. In the discussion section, we have added the following content (page 26; line 529-543):

“Mitochondria are pivotal organelles crucial for cellular metabolism. Disruptions in mitochondrial function have been linked to T Cell exhaustion, attributed to glycolytic reprogramming (66). Similarly, mitochondrial fragmentation has been closely associated with NK cell exhaustion (67).

However, the concept of NK cell exhaustion isn't as firmly established as it is for T cells. Exhausted NK cells should primarily exhibit diminished functions. This is characterized by a diminished ability to destroy tumor cells, a reduced capability to activate other components of the immune system, and compromised proliferation and survival rates. Additionally, this reduced functionality is associated with a decline in the expression of molecules responsible for cytotoxic activity, lower production of IFN-γ, and metabolic disturbances that may arise from mitochondrial dysfunction. While our current data is not sufficient to definitively classify these cells as exhausted NK cells, it supports that a subpopulation, referred to _Junb_hi cluster, demonstrates an exhaustion-like phenotype. The significant increase in this cell population following *Prdm1* knockout in NK cells may potentially be one of the reasons why *Prdm1ΔNcr1* mice lose their tumor-killing capacity. Whether the excessive expression of JunB in NK cells is also a contributing factor to their exhaustion, similar to T cells(65), requires further investigation.”.

Comment 49: Figure 7, x-axis labelling (MFI) of histograms is not correct. Do bar graphs and FACS plots show the same data? Does the number in the FACS plots indicate the MFI? If so, the FACS plots do not show representative samples?

Response 48: We appreciate the valuable comments provided by the reviewer. In the revised Figure 7, the MFI values have been removed. Bar graphs now display summary data from FACS histograms.

A representative sample close to the group's mean value was chosen for display in the histograms.

Comment 50: Figure 7, D - How are these data different from Figure 2H? Why is it now called "exhaustion", but not in 2H? Is the detected IFNg only driven by ex vivo stimulation with Il12/Il18? As above, a "standard" 4h assay should also be provided to allow better interpretation of potential differences. In the introduction, the authors cite the Ducimetiere study (Ref 5) highlighting "the primary function of ILC1 in suppressing the seeding of metastatic tumor cells in liver tissue". Thus, it would be interesting to test Ifng production by liver ILC1 and NK cells ex vivo at early time points of tumor inoculation.

Response 50: Tumors grow and proliferate within tissues, constituting one of the major causes of lymphocyte exhaustion. This part of the current study aims to investigate whether Prdm1 aids NK cells or ILC1 in resisting the exhaustion induced by malignant tumors. Specifically, we seek to ascertain whether the absence of Prdm1 renders NK cells or ILC1 more susceptible to exhaustion within the tumor microenvironment. Therefore, we will consider the capacity to secrete IFN-γ upon IL-12/IL-18 stimulation as one indicative aspect of exhaustion. It's crucial to emphasize that this assessment serves as only one piece of evidence, not the sole determinant. Overnight stimulation is a conventional method for studying NK cells and has been widely used across different laboratories, including our lab (e.g. Bream et al., *Blood*, 2003; Yu et al., *Immunity*, 2006; Wang et al., *J Clin Invest*, 2018). It's essential to clarify that our approach does not involve stimulating with tumor cells to evaluate the secretion capacity of IFN-γ by NK cells or ILC1.

**Reviewer 2 (Public Review):**
Summary:This study offers a significant advancement in understanding liver innate lymphoid cell (ILC) biology by elucidating the role of the transcription factor Prdm1. It shows that Prdm1 is crucial in maintaining the balance between conventional natural killer (cNK) cells and ILC1s in the liver, with knockout models revealing a vital role in cancer defense mechanisms. Despite not affecting direct cytotoxicity, Prdm1 deficiency leads to increased cancer metastasis and reduced secretion of key molecules like IFN-γ, pointing to its importance in immune regulation. The use of single-cell RNA sequencing further underscores Prdm1's role in cellular communication within the liver's immune milieu. This study is a robust contribution to the field, providing insights that could inform new immunotherapy approaches for liver cancer.Strengths:The study's strength lies in its comprehensive approach, combining the specificity of Prdm1 conditional deletion in Ncr1-cre mice with integrative omics analyses and cutting-edge cytometry to delineate Prdm1's role in liver Type 1 ILC biology and its functional implications in tumor immunity. This multifaceted strategy not only clarifies Prdm1's influence on ILC composition and maturation but also conveys potential therapeutic insights for liver cancer immunotherapy.

We sincerely appreciate your interest and critical assessment of our manuscript. We have carefully read your comments and suggestions, and I am truly grateful for your expert guidance. We have worked on addressing each of your concerns and comments, and below we provide a point-to-point response. Please find the detailed responses below:

WeaknessComment 1: A notable weakness of the study is the limited scope of in vivo disease models, primarily relying on the B16F10 melanoma model, which may not fully capture the complex behavior of Type 1 ILCs across diverse cancer types. Furthermore, the absence of direct human data, such as the effects of PRDM1 deletion in human NK cells or stem cells during their differentiation into NK and ILC1, leaves a gap in translating these findings to clinical settings.

Response 1: We appreciate the reviewer for raising these important points, which we see as a unique opportunity for future work to transform our understanding of Prdm1 and its targets as opposed to a weakness of the present study.

In our revised manuscript, we have discussed these limitations of our study (page 29; line 602-609):

“While our findings underscore the importance of Prdm1 in liver cNK cells and ILC1s tumor immune surveillance, it does not be validated in human NK cells, whereas previous studies have found that PRDM1 might inhibit the proliferation and function of human NK cells (33, 73). Furthermore, we not provided an in-depth evaluation in multiple tumor models. Further research may provide deeper insight into the role of PRDM1 in the anti-tumor function of human NK cells, enabling a more direct investigation of its application in cancer therapies. Given its important role in preserving liver cNK cells and ILC1s functional heterogeneity, enhancing Prdm1 function in human NK cells could potentially be a strategy to promote NK cell-based immunotherapy for cancer.”.

**Recommendations For The Authors:**
(Introduction)Comment 2: Reference 1 appears slightly misplaced. You might find the nomenclature discussion in Spits et al., Nature Reviews Immunology, 2013, more appropriate.

Response 2: We are really sorry for our inaccurate descriptions. According to Spits et al., (Spits et al., *Nature Reviews Immunology*, 2013) and other related studies, we have now adopted a more appropriate nomenclature as “Conventional NK cells” correspond to “cNK cells”, “Type 1 innate lymphoid cells” to “ILC1s”, and “Group 1 ILC” as the collective name of cNK and ILC1s.

The definition of these cells was described in the introduction (page 4, line 52-53; line58-62):

“Group 1 ILCs consist of cNK cells and ILC1s (1, 2), with distinct developmental trajectories and effect molecules (3).”, “In a state of homeostasis, liver group 1 ILCs (CD45+CD3-NK1.1+NKp46+) can be discriminated into cNK cells and ILC1s by the differential expression of CD49a and CD49b (2): cNK cells are marked by the expression of CD49b, while liver ILC1s exhibit a distinctive positivity for CD49a. Tumor Necrosis Factor Related Apoptosis Inducing Ligand (TRAIL) is also expressed on liver ILC1s, but not on cNK cells (10, 11).”.

We also describe cNK and ILC1 phenotypes in our scRNA-seq data, as shown in page 13; line 259-261:

“cNK cells expressed high levels of *Itga2* (CD49b) and Eomes, while ILC1s had high levels expression of *Itga1* (CD49a) and *Tnfsf10* (Supplemental Figure 5, F and G).”.

Comment 3: It has come to my attention that Reference 9 has been retracted. I recommend removing this citation to maintain the integrity of your references (https://doi.org/10.1182/blood.2023022801).

Response 3: We thank the reviewer’s comment and we now have removed this citation.

Comment 4: For a more comprehensive context around reference 15, consider citing Thierry Walzer's work ([https://rupress.org/jem/article/211/3/563/41636/T-bet-and-Eomes-instruct-thedevelopment-of-two]) which aligns closely with your discussion.

Response 4: We agree with the reviewer’s suggestion and have added this citation in our introduction (page 4; line 64-66):

“Liver environment facilitated T-bet expression in the early stage of NK cells development, which results in Eomes repression. The repression of T-bet is required for Eomes+ NK cells (17).”.

(Results)Comment 5: The NK cell signature referenced in 32 has been questioned for its reliability as discussed by Cursons et al., CRI 2019 (https://pubmed.ncbi.nlm.nih.gov/31088844/). Reanalysis of data in Figure 1 B/C and Supplementary Figure 1 with the refined NK cell signature from Curson's work would be advantageous.

Response 5: We thank the reviewer’s comment. As requested, we reanalyzed our data using the refined NK cell signature from Cursons et al. (revised Figure 1 A-C; revised Supplemental Figure 1). Of note, the overall survival of liver cancer (LIHC) patients only reached statistics significance when compared high and low expression of refined PRDM1-NK signature with a median cutoff (Figure 1, A-C). The overall survival performed with quartile high and low expression of refined PRDM1-NK signature was moved to supplemental figure 1, G-I.

The original text is: “Examination of 363 liver hepatocellular carcinoma (LIHC) patient samples from The Cancer Genome Atlas (TCGA) revealed a positive correlation between the expression of NK cell-associated genes (*NCR1*, *NCR3*, *KLRB1*, *CD160*, and *PRF1*) (32) and *PRDM1* expression (Figure 1A). Patients with top and bottom quartiles of *NK-PRDM1* signature expression were chosen for survival analysis (Figure 1B). Notably, patients with the _NK-PRDM1_hi signature had better overall survival compared to the these with NK-_PRDM1_lo signature (Figure 1C). Similar results were also found in skin cutaneous melanoma (SKCM, n=454) and lung adenocarcinoma (LUAD, n=497) patients (Supplemental Figure 1, A-F). These data suggested that *PRDM1* in NK cells might be essential for immune surveillance in some solid tumors, including liver cancer. These findings prompted us to investigate the impact and mechanism of *PRDM1* in NK cells and ILC1 within the context of liver cancer.”

We have rewritten this part in our revised manuscript (page 7; line 119-132):

“Examination of 363 liver hepatocellular carcinoma (LIHC) patient samples from The Cancer Genome Atlas (TCGA) revealed a positive correlation between the expression of NK cell-associated genes (34) (*NCR1*, *KLRB1*, *CD160*, *PRF1,* etc.) and *PRDM1* expression (Figure 1A). The patients are ordered from highest to lowest based on the expression of NK-Prdm1 for survival analysis (Figure 1B). Notably, patients exhibiting higher levels of NK-PRDM1 expression (above the median) experienced better survival outcomes compared to those with lower levels of NK-PRDM1 expression (below the median) (Figure 1C). Similar results were also found in skin cutaneous melanoma (SKCM, n=454) and lung adenocarcinoma (LUAD, n=497) patients (Supplemental Figure 1, A-F). Patients within the highest quartile of NK-PRDM1 signature expression demonstrated enhanced overall survival, a result that achieved statistical significance in LUAD and SKCM patients (Supplemental Figure 1, G-I). These data suggested that *PRDM1* in NK cells might be essential for immune surveillance in solid tumors, including liver cancer, and prompted us to investigate the function and mechanism of *PRDM1* in NK cells and ILC1 within the context of liver cancer.”.

Comment 6: The origin of the Ncr1-cre mice utilised should be clarified; is this the line developed by Eric Vivier? (https://www.pnas.org/doi/10.1073/pnas.1112064108).

Response 6: We did not use the line developed by Eric Vivier, our Ncr1-cre mice was purchase from Shanghai Model Organism Center, Inc. We described this in our method parts (page 29-30; line 612-614):

“*Prdm1fl/fl* mice were purchased from The Jackson Laboratory. *Ncr1-iCre* and *B2m-/-* mice were purchased from Shanghai Model Organisms Center, Inc. Six- to twelve-week-old littermates were used for the experiment.”

Comment 7: Considering the known reduction of Ncr1 expression in Ncr1-cre mice and its implications, it is recommended to repeat the B16F10 experiments with the correct control, Ncr1cre/+ Prdm1+/+.

Response 7: This is an excellent question, and it has been raised by another reviewer and comprehensively answered (Reviewer 1, Comment 1). The answer is below:

The expression of Cre and the insertion of loxP sequences both have the potential to influence gene expression. This is because the region where loxP is inserted may contain regulatory sequences for the gene of interest. Ncr1-Cre is a frequently used transgenic mouse model in our laboratory. In our prior research, we also had concerns about the possible impact of Cre on NKp46 expression, which could lead to a decline in NK cell function. Therefore, in our previous studies focused on Smad4 expression in NK cells, we conducted similar experiments. In Figure 6 of our published paper in the Journal of Clinical Investigation (Wang et al., *J Clin Invest*, 2018), we compared NKp46iCreTgfbr2fl/flSmad4fl/WT with NKp46-iCreTgfbr2fl/flSmad4fl/fl. Although the primary purpose is to establish Smad4's independence from TGF-β, it also allows for a comparison between Smad4fl/fl and Smad4fl/WT in the presence of Cre. In the critical phenotype we assessed, NKp46iCreTgfbr2fl/flSmad4fl/fl (compared with NKp46-iCreTgfbr2fl/flSmad4fl/WT) exhibited the same phenotype as NKp46-iCreSmad4fl/fl (compared with NKp46WTSmad4fl/fl). This suggests that Cre's influence on NK cells may be within a reasonable and controllable range. Furthermore, in contrast to the decrease in Ncr1 expression caused by Cre, the reduction in the expression levels of genes targeted by Loxp knockout, such as Prdm1 in this study (Figure 1 E), is more significant. Therefore, with the current techniques and research methods, we believe that the data provided in this study can support the role of Prdm1 in NK cells.

Comment 8: The proportion of ILC1 in wild-type mouse livers is notably higher than standard references. Could you confirm whether liver perfusion was performed before analysis? This procedure was not clearly detailed in the methods section.

Response 8: We apologize that we did not provide enough detail regarding this point in our original method. We had performed the liver perfusion before analysis. This has now been clarified in the method section of the revised text (page 30-31; line 630-636):

“Mice were perfused with 1◊ PBS by portal vein puncture before harvesting tissues. Liver and lung was digested with 0.05% collagenase II for 30 minutes and filtered through 70 µm cell strainers, and mononuclear cells were isolated after subjected to density gradient using 30% and 70% percoll. Spleen were also removed and pressed through 70 µm filterers to obtain splenocytes. Peripheral blood mononuclear cells were obtained from peripheral blood after lysis of red blood cells (Biolegend, 420301). Flushing femurs and mechanical disruption of inguinal lymph nodes were performed to obtain cells from bone marrow and lymph nodes.”.

The lymphocyte proportions in mice from different laboratories may exhibit slight variations, possibly due to genetic background disparities. To minimize the influence of genetic backgrounds, paired littermates were used in the current study, wherein one is Prdm1 WT and the other has the Prdm1 gene knocked out in NK cells.

Comment 9: There appears to be inconsistency in reference formatting; for instance, Ref 39 does not match the formatting of other references. A thorough review of your citation format is suggested.

Response 9: We apologize for the inadvertent errors and we reviewed the citation format.

Comment 10: The information in Figures 2B and C may be better suited to the supplementary section as it does not significantly contribute to the main text.

Response 10: We agree with the reviewer’s suggestion and these are now moved to supplementary figures (Supplemental Figure 2).

Comment 11: The citation of reference 40 could be strengthened by including Sathe et al., 2014, which directly pertains to your findings (https://www.nature.com/articles/ncomms5539).

Response 11: We added the suggested reference.

Comment 12: Can the findings presented in Figure 2D/F be replicated using alternative models?This would substantiate the versatility of your results.

Response 12: The current predominant in vivo tumor model for NK cells is primarily based on the use of B16F10 melanoma cells. These melanoma cells, with their low expression of MHC-I molecules, evade T cell-mediated immune surveillance, rendering them ideal targets for NK cells. Typically, this experimental melanoma metastasis assay involves tail vein injection, followed by nodules' detection in the lungs. To align with our investigation of liver-resident cNK and ILC1, we've introduced splenic injection (via the portal vein) and evaluated melanoma metastasis in the liver to reflect the anti-tumor capabilities of liver group 1 ILCs. We also explored subcutaneous tumor models, but we believe they may not effectively support Prdm1's role in cNK cells, particularly liver-resident NK cells and ILC1. While we've experimented with models using mouse liver tumor cells like Hepa 1-6, we found them less stable than B16F10 and less conducive to quantification. Should more suitable models or cells line emerge, we remain open to exploring them in future research.

Comment 13: The absence of in vitro killing assessments against B16F10 and YAC-1 leaves a gap in the NK cell characterisation which would be valuable to address.

Response 13: Isolating NK cells for ex vivo cytotoxicity assays typically requires stimulation with high concentrations of IL-2. Under such high IL-2 stimulation, many intracellular differences that contribute to difference in cytotoxicity, such as changes in transcription factors, are often masked. Another issue is that current ex vivo NK cell cytotoxicity assays often only isolate NK cells from the spleen. Liver-resident NK cells, on the other hand, are often limited in quantity and isolation methods, making it challenging to conduct ex vivo cytotoxicity assays effectively. If more sensitive detection methods become available, we will also incorporate ex vivo data into our future research endeavors.

Comment 14: The suggestion that NK cells produce IL-6 is indeed a bold one, and without additional validation through intracellular cytokine detection or ELISA, it may be prudent to omit these claims.

Response 14: We have checked the GSEA results, and found no valuable genes in IL-6 production.

Therefore, we have removed this figure.

Comment 15: The lack of fluorescence minus one (FMO) controls in Figure 3 and Supplementary

Figure 4 is noted; including these would enhance the validity of your gating strategies.

Response 15: As requested, we add the FMO controls in aforementioned figures.

Comment 16: There seems to be a minor mix-up in referring to Figure 4A in the scRNAseq results section, perhaps it was intended to refer to Figure 3A?

Response 16: We have corrected this part (line 247). We also double checked corrected the inaccuracies in the references to the figures. we apologize for the inadvertent errors.

Comment 17: The rich datasets generated from bulk and scRNAseq are commendable. However, I urge you to make these datasets publicly accessible with a GEO accession number.

Response 17: We appreciate the suggestion from the reviewer. We plan to upload our datasets when in the last version of our manuscript, which is also the request of the eLife policy.

Comment 18: Figure 4K is insightful, yet a similar analysis of the ILC1 cluster could provide a more rounded understanding.

Response 18: We thank the reviewer for the comments. We provide the similar analysis of ILC1s, as showing in revised Figure 5H.

Comment 19: The metabolic RNA signatures featured in Supplementary Figure 6 are intriguing and warrant further validation, perhaps through Seahorse analysis. Such validation could merit their inclusion in the main figures.

Response 19: This is a very good suggestion. Currently, our data offer only limited indications in this context. We have chosen to validate some aspects of Prmd1's influence on cytotoxicity molecules. As for Prdm1's impact on other aspects of NK cells, such as metabolic functions, we may explore further in future research. Additionally, we hope that by publishing our research findings, laboratories worldwide can draw insights for their own studies and conduct relevant research based on this data.

Comment 20: It is difficult to discern whether the cells depicted in Figure 7D are truly tumorinfiltrating ILC1 or NK cells that have adopted ILC1-like characteristics. Intravenous injection of CD45-PE could clarify this distinction, and if they are the latter, it may be more appropriate to refer to them as ILC1-like cells.

Response 20: We completely agree with the reviewer's suggestion that "tumor-infiltrating lymphocytes" may not be accurate for the current experiment. Therefore, in the revised manuscript, we have changed it to "liver cNK or ILC1 from tumor-bearing livers.